# celsr1a is essential for tissue homeostasis and onset of aging phenotypes in the zebrafish

Chunmei Li[1,2], Carrie Barton[3], Katrin Henke[1,2], Jake Daane[1,2], Stephen Treaster[1,2], Joana Caetano-Lopes[1,2], Robyn L Tanguay[3], Matthew P Harris[1,2]*

[1]Department of Genetics, Harvard Medical School, Boston, United States; [2]Department of Orthopedics, Boston Children's Hospital, Boston, United States; [3]Department of Environmental and Molecular Toxicology, Oregon State University, Sinnhuber Aquatic Research Laboratory, Corvallis, United States

**Abstract** The use of genetics has been invaluable in defining the complex mechanisms of aging and longevity. Zebrafish, while a prominent model for vertebrate development, have not been used systematically to address questions of how and why we age. In a mutagenesis screen focusing on late developmental phenotypes, we identified a new mutant that displays aging phenotypes at young adult stages. We find that the phenotypes are due to loss-of-function in the non-classical cadherin *celsr1a*. The premature aging is not associated with increased cellular senescence or telomere length but is a result of a failure to maintain progenitor cell populations. We show that *celsr1a* is essential for maintenance of stem cell progenitors in late stages. Caloric restriction can ameliorate *celsr1a* aging phenotypes. These data suggest that *celsr1a* function helps to mediate stem cell maintenance during maturation and homeostasis of tissues and thus regulates the onset or expressivity of aging phenotypes.

*For correspondence:
matthew.harris@childrens.harvard.edu

Competing interests: The authors declare that no competing interests exist.

## Introduction

Aging can be viewed as the progressive degeneration of tissue and physiological homeostasis through time. The regulation of aging is complex as it integrates environment, lifestyle, and genetic architecture to maintain homeostasis. In contrast, aging has a clear phylogenetic basis both in the maximum lifespan as well as the expressivity of aging traits. Evolution has shaped the manifestation of aging in different animals as is reflected in shared characteristics and mechanisms even between humans and yeast (*Bishop and Guarente, 2007*; *Kenyon, 2010*; *Vijg and Suh, 2005*). These common foundations have allowed the use of experimental laboratory and natural populations to investigate how and why we age. Unbiased genetic screens in Baker's yeast *Saccharomyces*, the nematode *Caenorhabditis elegans*, and the fruitfly *Drosophila melanogaster* have been instrumental in illuminating the genetic and biochemical aspects of aging of vertebrates. Similar unbiased approaches in the mouse have been limited, in part because of the restricted number of progeny produced as well as the fact that mice have relatively long lifespans, which restricts a systematic analysis of the mechanisms of aging from forward genetic approaches. Through the direct analysis of pathways identified in invertebrate and yeast models, however, the mouse has functioned as a key experimental system to identify shared aspects of aging and to understand modifiers of aging mechanisms through both environmental and genetic perturbations. Identification of alternative vertebrate models that can leverage the tools of forward genetics would be valuable to identify vertebrate specific regulators of this process.

Fish have long been important models in the study of aging and lifespan. In particular, guppies (*Poecilia reticulata*) have served as a natural and laboratory accessible model to address the causes

and evolutionary shaping of senescence (*Bronikowski and Promislow, 2005*; *Reznick et al., 2006*; *Reznick et al., 2004*). Guppies, however, are not well suited for forward genetic approaches leading to the need for other models for use in the laboratory. Recently, the killifish, *Nothobranchius furzeri*, has become an often used research organism to understand the causes of vertebrate aging (*Genade et al., 2005*; *Hu and Brunet, 2018*). The utility of this fish model has been due in part to their short lifespan, but like other teleost fish laboratory models they share experimental accessibility, allowing study of gene function (*Harel et al., 2015*; *Valenzano et al., 2011*). Although within strain variation in longevity and aging phenotypes are being addressed through genetic mapping (*Cui et al., 2019*; *Kirschner et al., 2012*; *Terzibasi et al., 2007*), to date this model has not been used in broader forward genetic approaches that have been the strength of prior work in other species to uncover how aging and longevity is encoded and can vary.

Zebrafish and medaka have been workhorse models for developmental genetics, however the use of these species to address aging has been limited (*Keller and Murtha, 2004*). Through reverse genetic approaches, studies have shown that zebrafish share telomere-mediated senescent programs and phenotypes of aging similar to that seen in other animals (*Anchelin et al., 2013*; *Carneiro et al., 2016*; *Henriques et al., 2013*). The phenotypic spectrum includes loss of tissue homeostasis, reduction in fecundity and fertility, arched spine or kyphosis, and shortened lifespan. Specific *lamin A* variants associated with aging-like phenotypes in Hutchinson Gilford progeria have been specifically tested in zebrafish and show analogous progeric aging phenotypes to human patients (*Koshimizu et al., 2011*). Similar loss-of-function experiments on medaka have not been carried out, although telomerase has been shown to be associated with senescent phenotypes (*Hatakeyama et al., 2008*; *Hatakeyama et al., 2016*), suggesting that this fish can support genetic analysis of senescence as well. These papers set the foundation for use of small laboratory fishes to study the genetic regulation of aging as they demonstrate shared phenotypic outcomes of known genetic regulators of aging. However, unlike invertebrate genetic models of aging, zebrafish and medaka are not particularly short lived, limiting efficient analysis of lifespan-extending changes. Leveraging the ability to process large numbers of larval zebrafish, Kishi et al. performed one of the first unbiased screens in zebrafish to identify genes associated with senescence using expression of Senescence-associate beta-galactosidase (SA-β-gal) as a biomarker (*Kishi et al., 2008*). This study is unique in approach, though specifically targets defects in tissue integrity observed in early larvae. As such, it remains unclear if these mutants are representative of the loci regulating normal aging.

Here, we report on a novel zebrafish mutant identified through a forward genetic screen for adult phenotypes that exhibits traits in early adulthood that closely resemble those associated with normal aging. The mutant does not show evidence of increased age-associated cellular senescence, but rather is deficient in maintaining tissue integrity through support of stem cell maintenance and proliferation. The phenotype is caused by loss-of-function mutations in the non-classical cadherin, *cadherin EGF LAG seven-pass G-type receptor 1a* (*celsr1a*). We observe a general loss of proliferative phenotypes in tissues suggesting that the progeric defect seen in mutants is associated with loss of homeostasis in adult tissues. Following we find that the function of *celsr1a* is necessary for the expression of stem cell factors in different tissues. These results suggest that *celsr1a* is linked to stem cell maintenance and/or proliferation and that disruption of its function leads to premature aging phenotypes in zebrafish. Affirming the role of *celsr1a* in aging programs, we show that caloric restriction can alleviate reduced viability and tissue level pathologies associated with *celsr1a* loss, in part through upregulation of *celsr1* paralogues. The identification of a zebrafish model for regulation of stem cell maintenance in aging opens up new avenues for aging research using zebrafish as a genetic tool for discovery.

## Results

### The identification of an adult zebrafish mutant with precocious geriatric phenotypes

In a large-scale screen for mutations affecting late development of the zebrafish, we isolated a class of mutants having altered scale patterning phenotypes and kyphosis in 10–12 week old adults (wpf, weeks post fertilization). These mutants displayed a broad collection of phenotypes that became more severe with age and resembled normal aging in zebrafish arising in our facilities in fish greater

than 18 months of age (*Figure 1*). The mutant phenotype was detected in both male and female fish. We focused on one of these mutants, named *fruehrentner (frnt)*, or 'early retiree' in German. The cumulative phenotypic effects from the *frnt* mutation lead to a progressive decrease in lifespan, with about half of mutant progeny dying before 9–10 months of age (*Figure 1D*). Fish living beyond this point showed progressive deterioration of their appearance and manifestation of sensorial neural defects causing them to swim erratically and in circles when presented with an acoustic stimulus (Suppl. *Videos 1* and *2*). Broadly these phenotypes resembled normal aging in wild-type zebrafish, however were apparent during early adult stages. Importantly, the *frnt* mutant exhibited no apparent outward morphological phenotypes as larvae or in juvenile stages. Instead, the observed phenotypes were acquired and only appear in early adult fish (*Figure 2*).

Histological analysis of adult *frnt* mutant zebrafish and wild-type controls showed clear defects in homeostasis of several tissues (*Figure 2*). In zebrafish, muscle fiber type is segregated in the trunk into a peripheral domain of slow muscle overlying fast muscle fibers (*Figure 2A*, *Figure 2—figure supplement 1A–B*). In *frnt*, the fibers of the slow muscle are severely affected and have smaller fiber size and hyperproliferation of mitochondria (*Figure 2C*, *Figure 2—figure supplement 1C–D*), many of which are degenerating (*Figure 2—figure supplement 1E–G*). Fast muscle fibers were not obviously affected in the mutant (data not shown). Histological analysis of aged wild-type fish shows comparable thinning of slow muscle fiber thickness as well as fibrosis of the surrounding tissue (*Figure 2B*). The *frnt* mutant also shows striking defects in the structure of the epidermis (*Figure 2D–F*). The epidermis of the adult zebrafish integument is a stratified epithelium with prominent cuboidal basal cells (*Figure 2D*). *frnt* mutants of comparable age show a drastic thinning of the epidermis with fewer basal cells and lengthened squamous cells overlying a thickened dermis (*Figure 2F*). A similar epidermal thinning and cellular structure is observed in old wild-type zebrafish (>2.5 years; *Figure 2E*). These results suggest that the *frnt* mutant affects tissues with high metabolic activity, such as the skin and slow muscle, and is reminiscent of phenotypes observed during normal aging in zebrafish.

The histological characteristics, such as sarcopenia and diminished basal cells of the epidermis, suggest that progenitor cell deficiencies may underlie these pathologies. Supporting this hypothesis, we found that expression of *pax7a*, a marker for muscle satellite cells (*Berberoglu et al., 2017*; *Seale et al., 2000*), was decreased in *frnt* slow muscle whereas analysis of a more general cell proliferation marker, *cdnk1a/p21,* did not show significant changes (*Figure 2G,H*). Following, we also

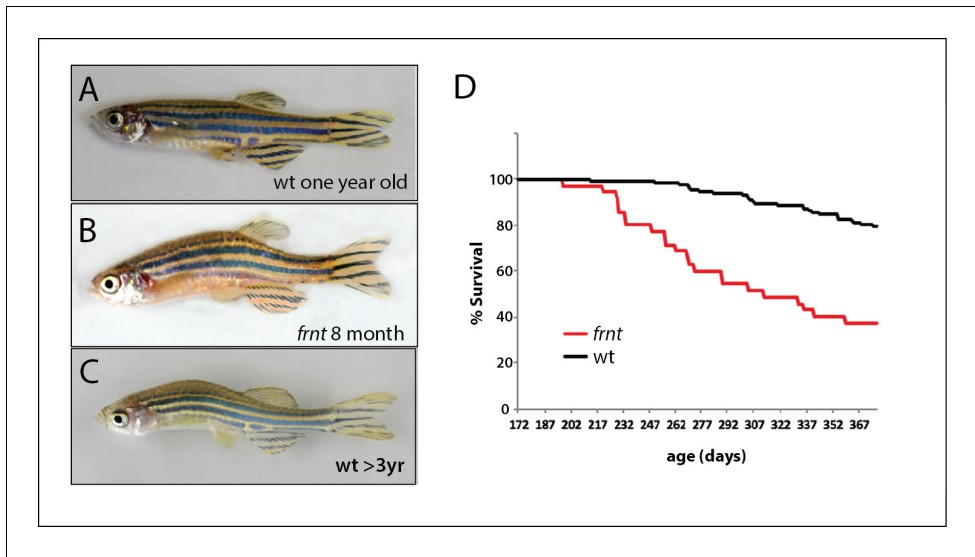

**Figure 1.** Identification of a zebrafish mutant, *fruehrentner* (*frnt*), exhibiting late phenotypes that resemble normal aging. (A–C) Aging phenotypes of young *frnt* mutant and old wild-type (wt) zebrafish. (B) Appearance of adult *frnt* mutant showing ruffled appearance and kyphosis at young adult stages closely resembles that of old fish (C) compared to a (A) wild-type fish of similar age. (D) *frnt* shows progressive loss of survival compared with wild-type fish.

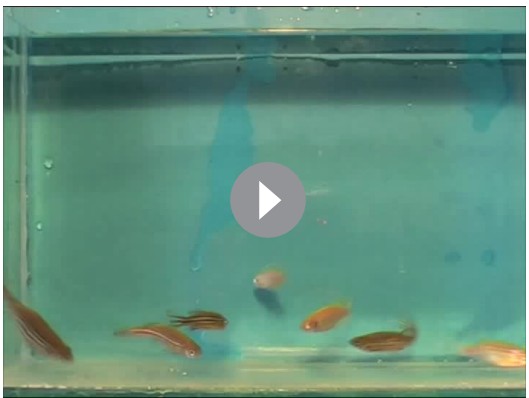

**Video 1.** Altered swimming behavior and response to acoustic stimuli in *celsr1a* mutants. *celsr1a* mutants (pigmented) and wild-type albino fish respond to periodic tank tap as an acoustic stimulus.
https://elifesciences.org/articles/50523#video1

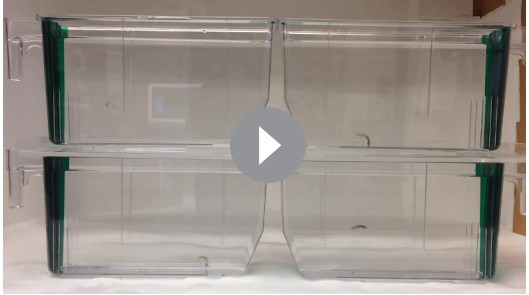

**Video 2.** Set up for behavioral analysis of *celsr1a* mutant fish after raising in diets of different caloric content.
https://elifesciences.org/articles/50523#video2

assessed expression of *ΔNp63*, which marks potential stem cells of the skin (*Guzman et al., 2013*; *Keyes et al., 2005*), as well as the epidermal tight junction marker *claudin-b* as a control in epidermis. We observed a similar decrease in expression in the *frnt* mutant specifically for *ΔNp63* but not *claudin-b* (*Figure 2J,K*). Thus, the acquired senescent phenotypes observed in the *frnt* mutant coincide with a decrease in progenitor cell markers in these tissues.

## The effect of *frnt* manifests late in development

We extended our analysis of the *frnt* mutant phenotype to ask when in development we were able to detect the onset of phenotypes observed in mature fish. Through histological analysis, we measured the development and maintenance of slow muscle through juvenile development. At 3 weeks of age, both *frnt* mutants and siblings have comparable slow muscle fiber diameter (*Figure 2I*). However, at 3 and 9 months of development, fiber size in *frnt* mutants is substantially smaller than in their wild-type siblings. This size difference is due to the decreased capacity of fibers to increase in size after 3 weeks of development (*Figure 2I*). Additionally, we used histological analysis of DAPI stained sections to identify changes in basal cell number in the developing zebrafish epidermis (*Figure 2L–N*). At 3 months of age there is little difference in basal cell number between *frnt* mutant and wild-type sibling controls. However, at 9 months *frnt* is deficient in the number of basal cells compared with age matched controls (*Figure 2L*). Thus, similar to slow muscle fibers, the *frnt* phenotype in skin is associated with a failure to increase in cell number. These results suggest that the *frnt* phenotypes manifest during late development, increase in severity with progressive age and affect proliferative/growth potential of maturing tissues.

## The *frnt* phenotype does not stem from increased senescence

Cellular senescence is thought to be one factor regulating homeostasis and onset of aging within tissues (*Collado et al., 2007*). Hallmark phenotypes of senescence are loss of telomere length as well as activity of lysosomal β-galactosidase, commonly referred to as senescence-associated β-galactosidase (SA-β-gal) (*Lee et al., 2006*; *Dimri et al., 1995*). Telomeres act as essential regulators of genomic stability that allow for fidelity in genome replication. In each replication of chromosomes, telomere length is maintained by a specialized molecular complex, shelterin, through action of the *tert* gene product.

To understand if senescence was an underlying basis of the *frnt* phenotype, we first looked at total telomere length in mutant zebrafish tissue by Southern blot (*Figure 3*). For a positive control, we analyzed telomere length in first generation *tert* homozygous mutants, as these mutants have been shown to exhibit late age-related phenotypes and accumulation of senescent biomarkers (*Anchelin et al., 2013*; *Henriques et al., 2013*). Southern blots from 1 year-old *tert* mutant tissues show a distinct reduction of average telomere length. In contrast, age matched *frnt* mutants do not show an appreciable change compared to wild-type fish (*Figure 3A*). We further analyzed the

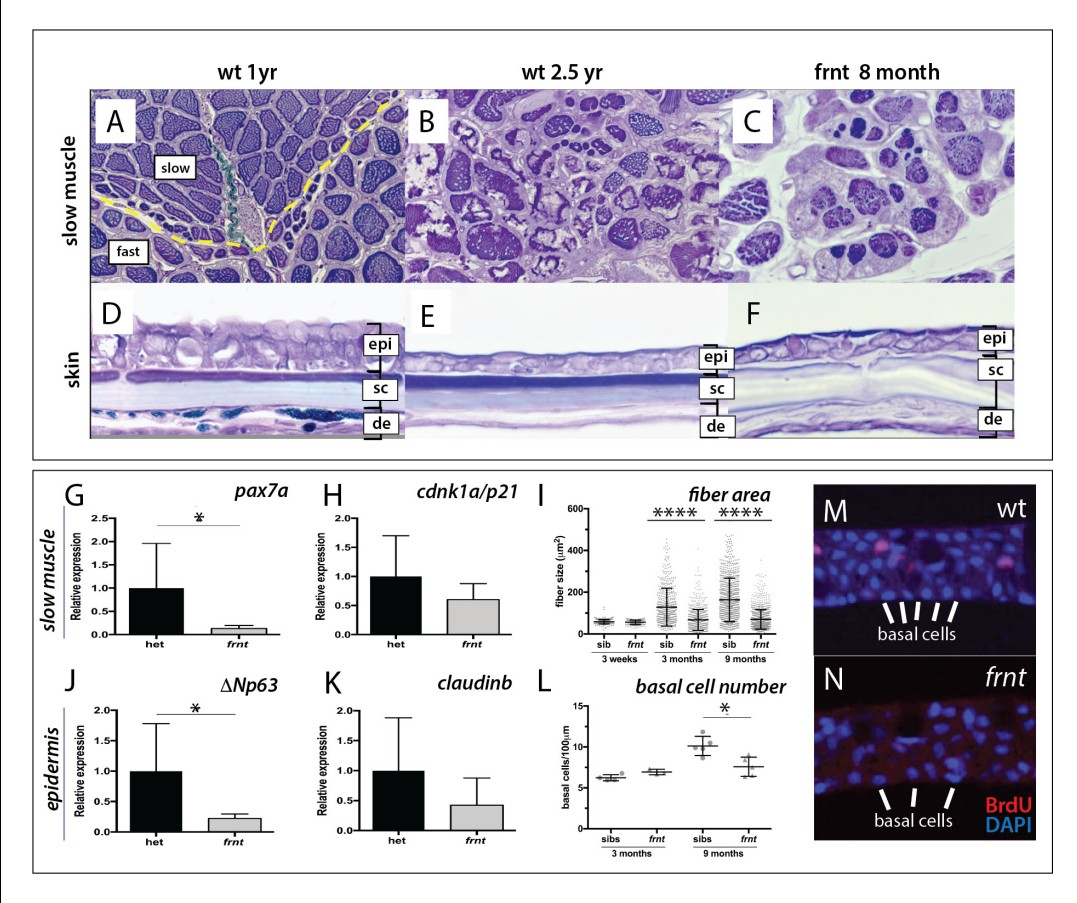

**Figure 2.** *frnt* causes acquired deficiencies resembling normal aging. (A–F) Direct comparison of *frnt* with young adult, (A, D) and naturally aging wild-type zebrafish (2.5 years; (B, E). Similar degenerative pathologies are shared between aged zebrafish (B, E) and 8 month-old mutants (C, F), such as fibrosis and sarcopenia of slow muscle fibers (B, C), and thinning skin (E, F). (G–I) Characterization of slow muscle phenotype in *frnt* mutants. (G) Expression analysis of stem cell marker *paired-box 7a* (*pax7a*, n = 6) and (H) *cdnk1a/p21* control in slow muscle from 7 month old homozygous (n = 6) and heterozygous mutant fish (n = 9). (I) Adult *frnt* has smaller fiber size in slow muscle compared to age-matched wild type and sibling fish (3 month (n = 5) and 9 month old (n = 9), but not as juvenile fish (3 week old, n = 3). (J–L) Changes in epidermal phenotype in *frnt* mutants. (J) Expression of the stem cell marker *delta-Np63* and (K) control *claudin-b* in epidermal tissues from 7 month old homozygous (n = 12) and heterozygous (n = 5–6) *frnt* fish. (L) Count of DAPI positive basal cells in the integumentary epithelium in 3 (n = 3–4) and 9 month old (n = 5) *frnt* fish. (M–N). DAPI stained epidermal nuclei in wild-type (M) and *frnt* (N) mutant epidermis. Error represented as mean +/- standard deviation. ****p<0.0001, *p<0.05.

The online version of this article includes the following figure supplement(s) for figure 2:

**Figure supplement 1.** *celsr1a* affects mitochondrial proliferation and maintenance.

---

activity of SA-β-gal in histological sections of *frnt*, *tert* mutants and age-matched wild-type tissues as a measure of senescence. Compared to heterozygous siblings, *tert* homozygous mutants show considerable activity of SA-β-gal (*Figure 3B,C*). In contrast, we saw no discernable difference between *frnt* homozygous mutants and wild-type controls (*Figure 3D,E*). Thus, there is little evidence that the *frnt* phenotype is due to activated senescence programs typically observed in *tert* deficiencies.

## Identification of the genetic cause of *frnt* aging phenotypes

To identify the genetic locus affected in the *frnt* mutant, we used whole genome sequencing and mapping based on homozygosity-by-descent (*Bowen et al., 2012*). Initial mapping showed tight linkage to chromosome 4 (*Figure 4A*). Efforts to refine the map interval using polymorphic markers was limited as the linked region fell within a large chromosomal interval showing low heterozygosity

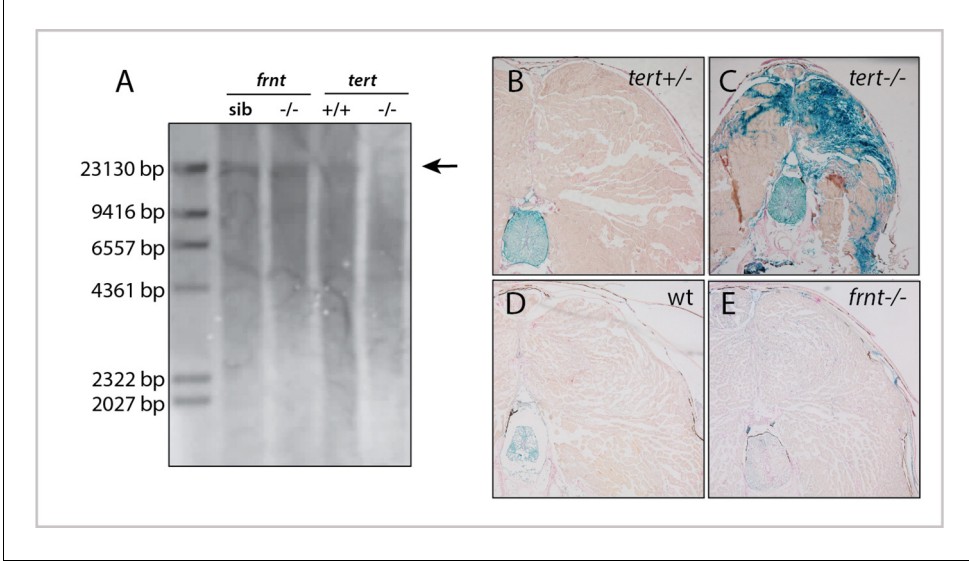

**Figure 3.** *frnt* mutation does not affect senescence biomarkers. (**A**) Southern blot of genomic DNA of zebrafish *telomerase reverse transcriptase* (*tert*) and *frnt* mutants probed for telomere repeats. Data shows no effect of the mutation on telomere length in the *frnt* mutant compared with significant decrease of intensity and size of telomere in *tert* deficient fish. Arrow points to upper size expected for zebrafish telomeres; n = 4 for each group. (**B–E**). Senescence-associated beta-galactosidase (SA-βgal) staining (blue) of *frnt* mutants. (**B, C**) Positive signal of senescence as shown in 5 month old *tert* mutants (**C**), n = 3 compared with their heterozygous controls (**B**), n = 3. Comparably staged wild-type and *frnt* adults (**E**), n = 4 show resting levels of SA-βgal signal. Samples were counterstained with nuclear red.

and limited recombination (*Figure 4C,D*). As we had genomic sequence of the whole interval, we were able to define several missense mutations as potential candidate mutations for causing the *frnt* mutant phenotype, however, there were too many mutations to functionally address. Thus, we performed a non-complementation screen to identify further alleles of *frnt* to define the affected gene. First, using N-ethyl-N-nitrosourea (ENU) induced mutagenesis of wild-type zebrafish, we identified a mutant (*mh36*) within progeny from crosses to *frnt* homozygous fish that failed to complement *frnt*. Sequencing the exome of homozygous *mh36* led to the identification of a nonsense mutation (C1693X) in the gene *celsr1a* within the linked interval of *frnt* (*Figure 4E*). As chemical mutagenesis can lead to many mutations and the mapping interval was large, we extended this approach by making targeted deletions using CRISPR/Cas9 mediated gene editing in the *frnt* heterozygous background. We were successful in identifying mutants that exhibited the *frnt* aging phenotype having insertion/deletions predicted to lead to premature truncation of the *celsr1a* gene product (*Figure 4F*). Following these non-complementation approaches, we reassessed our mapping in the original *frnt* mutant. In depth analysis of the whole genome sequence data of *frnt* mutants at the *celsr1a* gene locus uncovered a unique transposon insertion of approximately 3.5 kb into exon 1 of the *celsr1a* (*Figure 4G*). The effect of this insertion is predicted to lead to an early truncation of the protein. The identification of a transposon as a cause of the *frnt* phenotype, suggested that the allele was present in the background founders used in the screen. Supporting this conclusion, we found that several isolated families having this phenotype were derived from the same original male founders. Thus, through our mapping of *frnt* and non-complementation analysis, we have identified that the *frnt* phenotype is due to a disruption of *celsr1a* function.

*celsr1a* is an atypical cadherin of the flamingo family of cadherins. Among vertebrates, there are three ancestral orthologues that are shared, Celsr1-3. Fish have two orthologues of *celsr1*, *celsr1a* and *celsr1b* (*Formstone and Mason, 2005*) stemming from a whole genome duplication shared among teleosts. Celsr1a is a large membrane bound protein extending greater than 3000 amino acids in length. The mutations identified all lie in the N-terminal extracellular domain. Given that the mutations cause premature truncations or frameshifts upstream of the first transmembrane domain (*Figure 4H*), we predict that the *frnt* phenotype is due to loss of *celsr1a* function. The identified

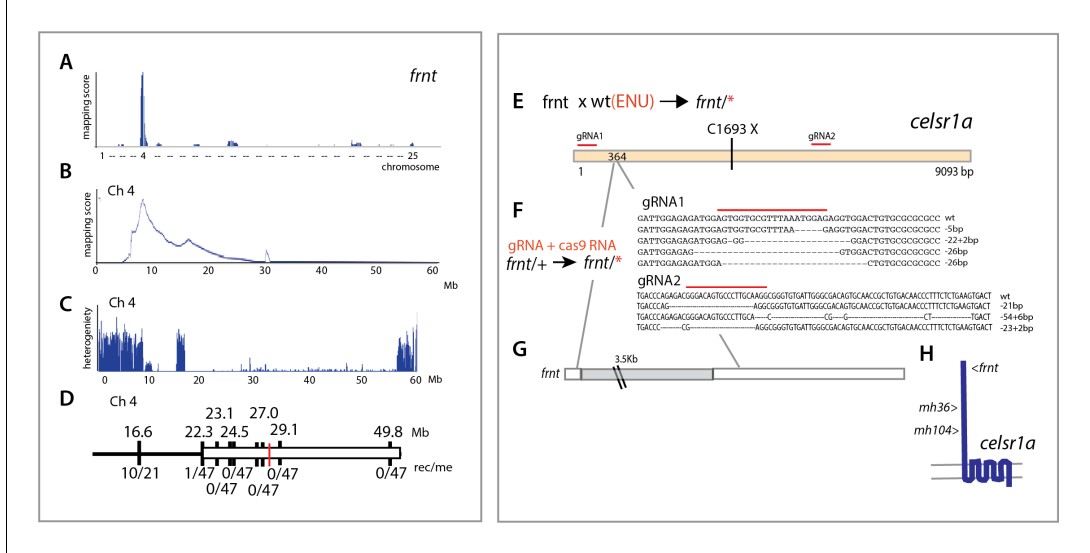

**Figure 4.** Identification of altered *celsr1a* function underlying the *frnt* phenotype. (**A**) Mapping by homozygosity-by-descent indicates linkage of *frnt* to chromosome 4. (**B**) Mapping score across chromosome 4. (**C**) Analysis of heterogeneity across chromosome four showing a broad region of homogeneity in the population indicating linkage. (**D**) Fine mapping of *frnt* showing limited recombination and resolution of the map position along chromosome 4; *white bar*, area showing linkage; red hashmark, position of *celsr1a*; top, position (megabase, Mb) on chromosome 4, zv9 assembly (https://ensembl.org); bottom number of recombinants per meiosis (rec/me) scored. (**E**) Chemical mutagenesis loss-of-complementation screen to identify the gene mutation underlying the *frnt* phenotype. Exome sequencing of identified founders having the *frnt* phenotype (*frnt/\**), identified mutations in the *celsr1a* gene within the mapped interval (*mh36, C1693X*). (**F**) Identified deletions/insertions within *celsr1a* generated through CRISPR/Cas9 genome editing that fail to complement *frnt*. Recovered sequences from F1 founders; guideRNA position demarcated with overlain red bar. Of the recovered lines, allele *mh104_P2027A-fs11X* was retained **G**) Identification of transposon insertion within *celsr1a* in *frnt*. **H**) Schematic of *celsr1a* and position of identified mutations.

alleles all have comparable phenotypes and fail to complement each other, supporting the identified mutants as *celsr1a* nulls.

## *celsr1a* expression wanes with age

Analysis by whole mount *in situ* hybridization has previously shown *celsr1a* to be broadly expressed during gastrulation and early larval development (*Carreira-Barbosa et al., 2009*; *Formstone and Mason, 2005*; *Harty et al., 2015*) (https://zfin.org). To assess differential expression of *celsr1a* during development, we used CRISPR/Cas9-mediated homology directed repair to knock-in a *green florescent protein (GFP)* coding sequence into the endogenous *celsr1a* locus. We isolated an expressing line with insertion of GFP 114 nucleotides upstream of the translation initiation site in the 5' UTR of *celsr1a* (*Figure 5A*). As the insertion allele fails to complement *frnt,* we predict that the allele is disruptive of normal *celsr1a* regulation and function. The identified line, *celsr1a^GFP*, recapitulates early expression seen by whole mount *in situ* (1dpf, *Figure 5B–C*), and strongly labels the eye, the central nervous system, the lateral line, the mesonephros and the intestine in young larvae (4dpf, *Figure 5D*). On close inspection, *celsr1a^GFP* is expressed, albeit at lower levels, in both epidermis and slow muscle (*Figure 5E–F,H*). These tissues show strong pathologies in the mutants (*Figure 2*). Notably, only select cells are labeled in the early epidermis, suggesting differential expression of *celsr1a* within this tissue (*Figure 5E*). At 12dpf, *celsr1a* expression remains prominent in slow muscle fibers, (*Figure 5H* and data not shown) and in the intestinal epithelium (*Figure 5I,J*). Similar to findings by Hardy et al, (*Harty et al., 2015*), we find that *celsr1a* expression wanes in late development. Expression of *celsr1a* in adults is retained primarily in neuromasts and with a low expression level throughout other tissues (*Figure 5K–M*). However, signal was retained in specific cells in different tissues as shown in localized cells in the intestine (*Figure 5L,M*).

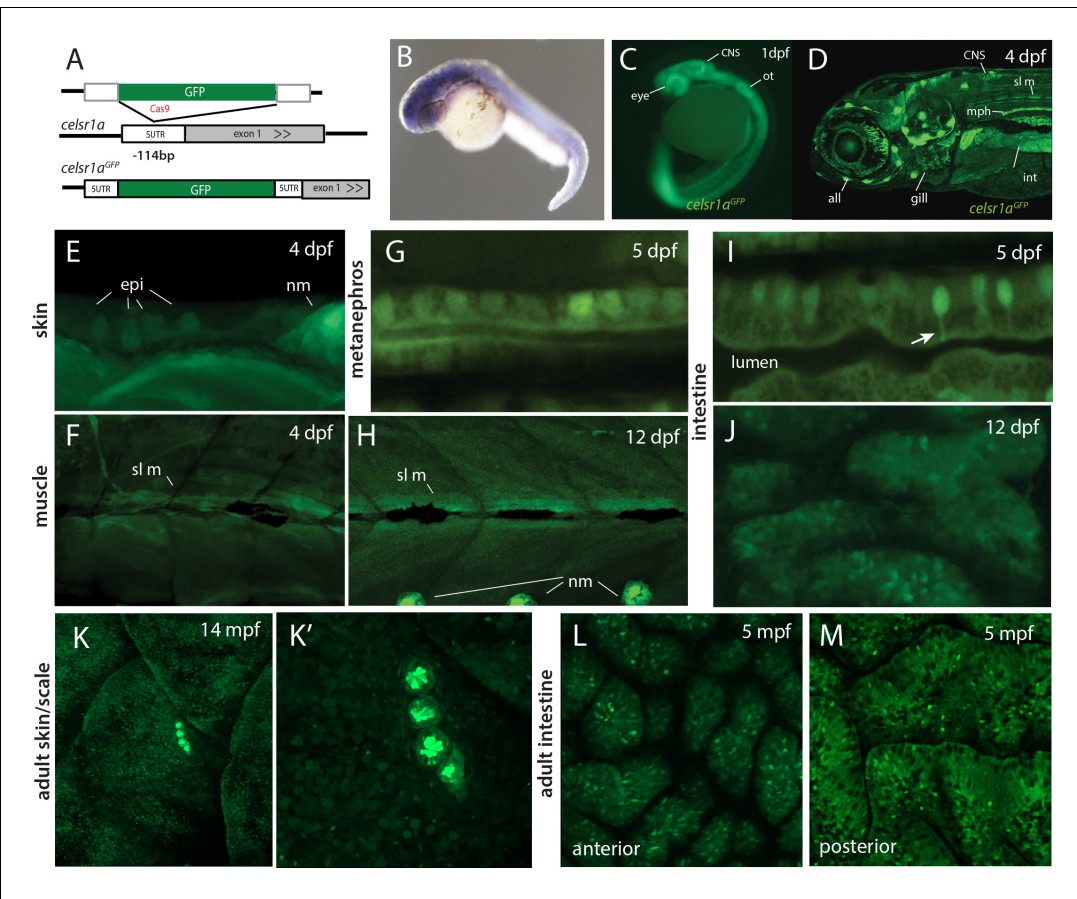

**Figure 5.** *celsr1a* expression during development and as a function of age in adults. (**A**) Strategy for GFP insertion at the *celsr1a* endogenous locus by homology directed repair. (**B**) Whole mount *in situ* analysis of *celsr1a* expression in early development (1dpf). (**C**) Expression of *celsr1a* in central nervous system (CNS), peripheral nervous system (PNS), and gut in the *celsr1a*^GFP/+ transgenic line. (**D–J**) Expression of *celsr1a*^GFP/+ transgene in larval, and juvenile zebrafish showing progressive restriction and localization to specific cell types and tissues. *all*, anterior lateral line; *epi*, epidermal cells; *int*, intestine; *mph*, metanephros; *nm*, neuromast; *sl m*, slow muscle; *ot*, otolith. (**K–M**) Restricted expression of *celsr1a*^GFP in adult tissues. (**K**) Expression of *celsr1a*^GFP in adult skin and neuromasts covering the lateral surface. (**L–M**) Expression of *celsr1a* is shown in anterior and posterior regions of the adult intestine.

## *celsr1a* is required for polarity of integumentary appendages

Celsr1 is a key component of planar cell polarity (PCP). In concert with Frizzled, Van Gogh (Vang), and non-canonical Wnt signaling factors, Celsr1 regulates cell asymmetry and developmental signaling pathways (*Goffinet and Tissir, 2017*; *Tissir and Goffinet, 2013*). The role for the other orthologues, Celsr2 and 3 is unclear. A Celsr1-deficient mouse has been used to study the role of PCP in development. Homozygous *Celsr1* mice show pelage phenotypes with misaligned hair follicles and the appearance of whirls (*Devenport and Fuchs, 2008*; *Ravni et al., 2009*). A similar phenotype is also seen in the patterned arrays of tongue papillae in *Celsr1* deficient mice (*Wang et al., 2016*). Both phenotypes are also observed in mice with alterations in *Vang2* gene function (*Devenport and Fuchs, 2008*; *Wang et al., 2016*) and are considered reliable readouts of PCP signaling in adult mice.

Analogous structures to hair of mammals in zebrafish are scales. In contrast to hair, which is primarily an ectodermal derivative, scales in fishes are primarily mesodermal, comprising components of the dermal skeleton. However, scale development is dependent on the formation of an ectodermal placode, a structure homologous to the placodes necessary for other integumentary structures

such as hair and feathers (*Harris et al., 2008*). Thus, early aspects of formation and patterning are conserved between divergent structures of scales and hair, and it has recently been shown that further downstream patterning is similar as well (*Aman et al., 2018*). Scales form ordered arrays of overlapping surface skeletal elements across the body (*Figure 6A*). Within each scale there is an internal polarity, biasing the growth to the caudal aspect of the scale from an initial osteogenic focus (*Figure 6C*); this polarized accretionary growth leads to the formation of overlapping arrays of scales along the flank of the fish. *Iwasaki et al. (2018)* have demonstrated that scale patterning is sensitive to PCP signaling in the ectoderm, resulting in formation of scale 'whirls' comparable to those seen in mice with altered *Celsr1*. We investigated scale formation in the zebrafish as biomarkers of altered PCP signaling in *frnt* mutants. At the earliest timepoints of scale development analyzed, *frnt* mutants had obvious scale patterning defects (~8 wpf). These defects are maintained in adults showing spiraling patterns of scales on the flank (*Figure 6B*). Furthermore, individual scales in *frnt* mutants show radial patterning in stark contrast with the polarized growth of scales from wild-type individuals (*Figure 6C–E*). Thus, alteration in *celsr1a* in zebrafish affects the patterning of structures analogous to hair follicles affected in the *Celsr1* mouse mutant. These phenotypes are consistent with a role of *celsr1a* in PCP signaling during zebrafish development.

## *celsr1a* is required for proliferative capacity and maintenance of intestinal progenitor cells

One of the more consistent phenotypes in aging is loss of tissue organization and homeostasis as a function of age. Our histological analyses suggest that several tissues in *celsr1a/frnt* mutants are diminished associated with decreased expression of stem cell markers (*Figure 2*). The intestinal epithelium has served as a fundamental model of how stem cells within a tissue are specified and maintained. However, only few papers have detailed differentiation and stem cell biology and differentiation in the intestine in fishes (*Aghaallaei et al., 2016*; *Crosnier et al., 2005*; *Li et al., 2019b*; *Lickwar et al., 2017*; *Wallace et al., 2005*; *Zhao and Pack, 2017*). To investigate the role of *celsr1a* in maintaining tissue homeostasis and progenitor populations in adult tissues, we analyzed changes in the intestinal epithelium in the *frnt* mutant. Consistent with our findings in other tissues in the mutant, analysis of the histological pathology of the intestine of *frnt* demonstrates a significant decrease in epithelial thickness and a reduction of the anterior gut circumference (*Figure 7—figure supplement 1A–D*).

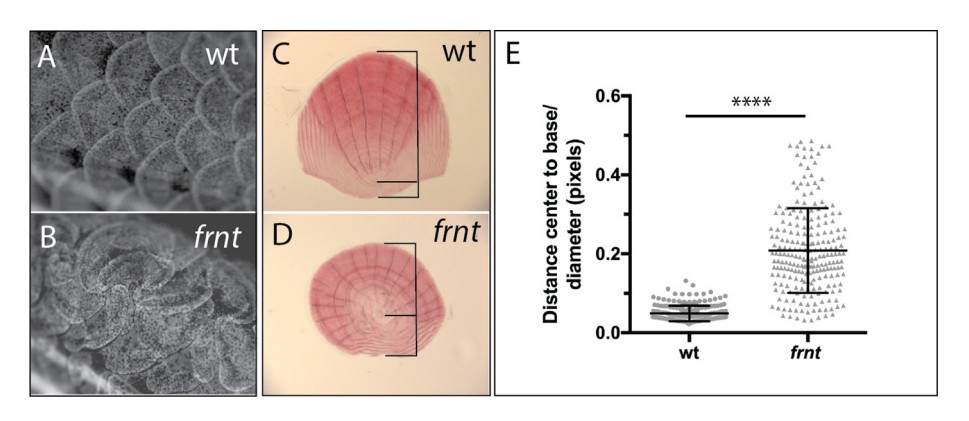

**Figure 6.** Loss of *celsr1a* leads to altered polarity phenotypes in the zebrafish integument. (**A–B**) Alizarin red stained adult wild-type zebrafish showing pattern of scales on the flank. (**A**). Wild-type fish showing normal, regularly spaced pattern of scales whereas, in (**B**) *frnt* mutants show altered patterning of scales across the flank, creating swirls. (**C**) Wild-type scale showing internal polarity of growth along the rostrocaudal axis (bottom to top). (**D**) *frnt* scale showing radial pattern of growth rings (annuli), without the internal polarity normally observed in wild-type. (**E**) Quantification of polarity in wild-type sibling and *frnt* scales (n = 6 fish,>25 scales each side) as indicated by the ratio of the center-to-base normalized by the diameter for each scale (**C, D**). Data presented as mean ± standard deviation; ****p<0.001.

We find that the mutant phenotype in the intestine is associated with drastic changes in the proliferative capacity of the intestinal epithelium. After short term Bromodeoxyuridine (BrdU) labeling, adult *frnt* mutants showed negligible BrdU incorporation in the intestine compared to age matched controls (*Figure 7A–D*, *Figure 7—figure supplement 2A*). Identified BrdU positive cells were found localized near the base of rugae. Consistent with these findings we show reduction of phospho-histone H3 labeling of mitotic cells in rugae (*Figure 7—figure supplement 2D*). BrdU incorporation in intestinal epithelium of larvae in which *celsr1a* cells are marked with GFP (*celsr1a^{GFP}*) shows restricted incorporation of BrdU in *celsr1a^+* cells during growth (*Figure 7J–K*). This suggests that *celsr1a*-expressing cells in the larval intestine are not actively cycling. In an effort to address maintenance of progenitor pools in the intestinal epithelium, whole mount *in situ* analysis of potential stem cell markers, such as *sex determining region Y-box 2* (*sox2*) (*Kuzmichev et al., 2012*; *Que et al.,*

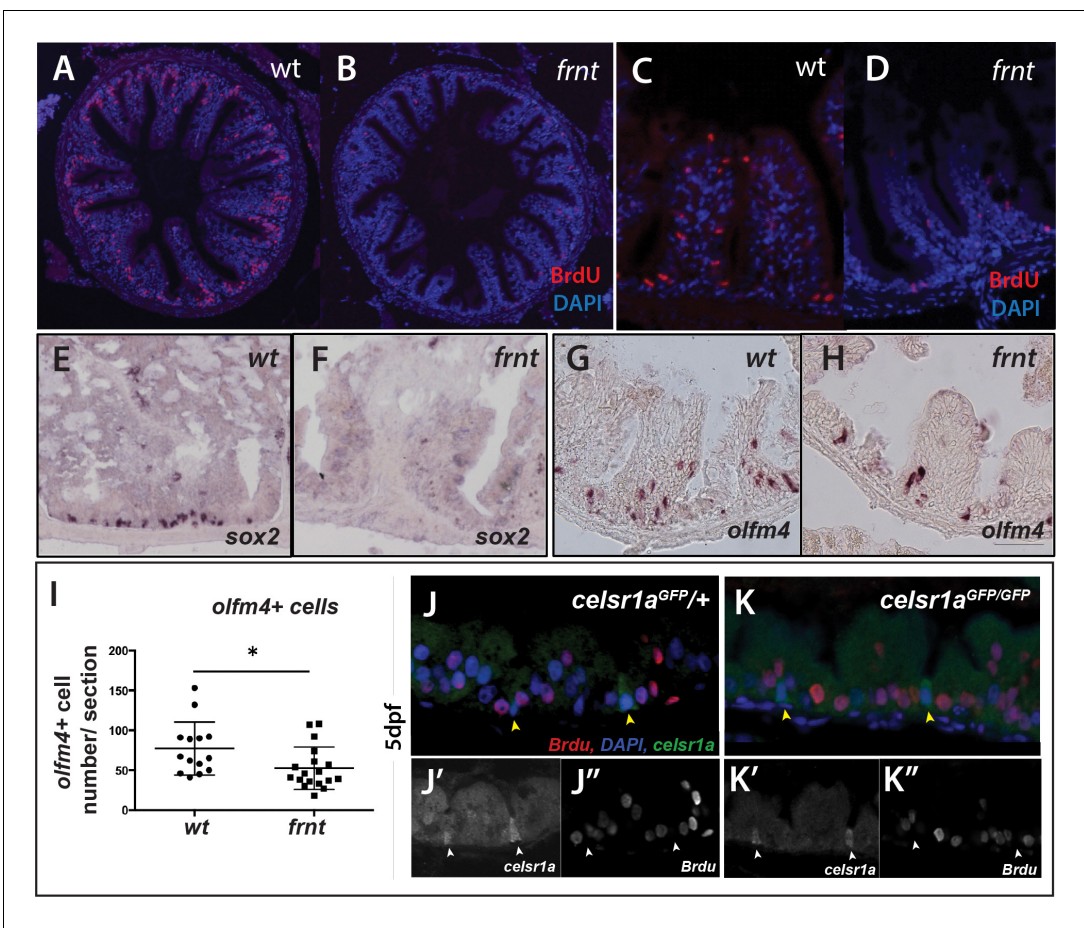

**Figure 7.** *celsr1a* is essential for activity and maintenance of progenitor cells. (A–D) Analysis of proliferative capacity of the adult intestine in *celsr1a* mutants and age matched wild-type fish (pulse injection and incorporation after 4 hr (red), nuclei counterstained by DAPI. (A, B) Low power view of comparable posterior regions of intestine of wild-type (A) and age and size matched *frnt* mutants (B). (C, D) Close up of intestinal rugae showing cells incorporating BrdU. (E–H), *in situ* hybridization of expression of *sex-determining region Y-box 2* (*sox2*) (E, F) and *olfactomedin 4* (*olfm4*). (G, H) genese in adult intestinal epithelia of wild-type (E, G) and *celsr1a* mutant (F, H) zebrafish. (I) Quantitation of changes in the number of *olfm4^+* cells observed in mutants; data presented as mean +/- standard deviation, *p<0.05, n = 5 (wt sibling), n = 7 (*frnt*) (J–K) Proliferative cells (24 hr after BrdU pulse, red) in comparison to *celsr1a* expression (*green*, yellow arrowhead) in larval developing intestine at 5dpf in (J) *ceslr1a^{GFP}* heterozytotes and (K) *ceslr1a^{GFP}* mutants; DAPI, blue. Insets J' and J'' and K' and K''are separate channels showing *celsr1a* expression and Brdu detection, respectively in the same tisse.

The online version of this article includes the following figure supplement(s) for figure 7:

**Figure supplement 1.** c*elsr1a* affects intestine growth and homeostasis.

**Figure supplement 2.** Cell cycling is diminished in *celsr1a* mutant intestinal epithelia.

*2007*; *Chen et al., 2015*; *Figure 7E,F*) and *olfactomedin 4* (*olfm4*) (*Figure 7G–I*; *van der Flier et al., 2009*; *Igarashi and Guarente, 2016*), were performed in the adult intestine of wild-type and *frnt* mutant fish. Supporting our expression analysis in skin and slow muscle, we detected a strong reduction in *sox2* and *olfm4* positive cells in *celsr1a* mutants, suggesting that *celsr1a* is required for normal maintenance of progenitor cells in the intestinal epithelium.

One hallmark of resting stem cells is their slow cycling during normal tissue homeostasis. To further determine the effect of loss of *celsr1a* function on proliferative capacity, we assessed the retention of BrdU at extended chase periods to permit detection of slower cycling cells. Analysis of single nucleoside dosing events over 48 hr indicated a progressive reduction of differences between mutants and siblings in the cells retaining or incorporating BrdU label in the intestine (*Figure 7—figure supplement 2*). These data suggest that existing progenitor populations are retained in the *celsr1a* mutant and are able to proliferate at these late stages in a limited capacity.

*celsr1a*^GFP^ is expressed in a subset of cells in the developing intestinal epithelium (*Figure 5*) resembling enteroendocrine cell (EEC) morphology. *Neuronal differentiation 1* (*neurod1*), is a transcription factor associated with notch signaling, which is a late marker for EECs in the intestine of mice and zebrafish (*Li et al., 2011*; *Lickwar et al., 2017*). Using the transgenic line, *Tg(neurod1:TagRFP)*, we found that in early development, the transgene labels a subset of *celsr1a*^GFP^ positive cells (*Figure 8A–C*), suggesting that the function of *celsr1a* may predominate in EECs. At this stage a number of *celsr1a*-expressing cells were identified without *neurod1* expression, suggesting that *celsr1a* represents an earlier stage in their specification. The larval intestines of the mutant are markedly thinner however retain a complement of *celsr1a*-expressing cells (*Figure 8A–C*). Quantitation of overlap was not feasible at early stages due to low signal intensity. Analysis of adult intestines

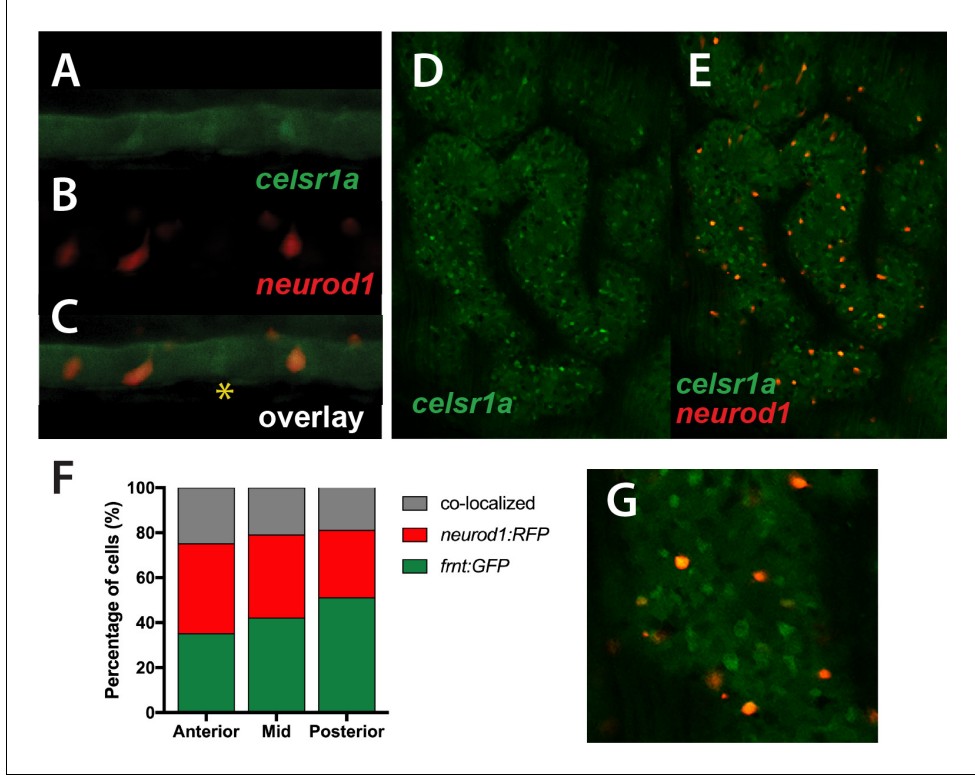

**Figure 8.** Colocalization of *celsr1a* and *neurod1* positive sensory enteroendocrine cells during development. (**A–C**) Co-expression of *celsr1a* (*celsr1a*^GFP/+^) and *neurod1* (*Tg(neurod1:TagRFP)*) in four dpf intestine; asterisk highlights *celsr1a*^+^ without *neurod1* expression; all pictures luminal side placed on top. (**C**) Overlay, shows predominant colocalization early in this cell population. Expression was seen in all larvae analyzed (n = 6). (**D–E**) Expression of *Tg (celsr1GFP)* and *neurod1* in 5 month old adult zebrafish intestine. Area pictured is from the middle intestinal region. (**F**) Quantitation of expression and overlap between markers (n = 6, 1–4 pictures/fish/region). (**G**) Close up of overlay showing distinct marked cell populations.

show significant co-expression of *celsr1a*$^{GFP+}$ and *Tg(neurod1:TagRFP)*$^+$ cells however, retaining individually expressed cell populations (*Figure 8D–G*).

## Effect of caloric restriction on *celsr1a* phenotypes

The phenotypes we observed in *celsr1a* deficient animals resemble the anatomical and behavioral aspects of normal aging in wild-type zebrafish. To assess the role of *celsr1a* in mediating aging processes, we wanted to analyze how alteration in mechanisms previously related with progression of aging phenotypes would affect *celsr1a* mutant phenotypes. Caloric and dietary restriction are two commonly used strategies that have been shown across animals to have a consistent protective effect on the manifestation of aging phenotypes (*Fontana and Partridge, 2015*; *Speakman and Mitchell, 2011*). The regulation of these effects on reducing aging phenotypes is thought to be in part through the action of the Sirtuin family of acetyltransferases (*Guarente, 2013*), mTOR (*Blagosklonny, 2010*) and Insulin receptor/Foxo signaling (*Kim et al., 2015*; *Mouchiroud et al., 2013*). In fish models, the effects of dietary restriction on aging and age-related pathologies have been mainly tested in zebrafish, in which most studies use overall dietary restriction as means of nutritional regulation (*Adams and Kafaligonul, 2018*; *Arslan-Ergul et al., 2016*; *Novak et al., 2005*). Such treatment regimens have shown changes in age-related neurological and behavioral phenotypes (*Adams and Kafaligonul, 2018*) and can have long term impacts on maintenance of weight and health of the fish (*Arslan-Ergul et al., 2016*). Caloric restriction (CR) regimens have been tested in zebrafish, however the outcomes on age-related phenotypes have not been reported (*Robison et al., 2008*). Although dietary restriction has the potential to alleviate age-related phenotypes, the extent by which this regulation operates in fishes remains an open question.

We set out to test if modulation of caloric restriction would attenuate the pathology observed in *celsr1a* mutant fish. Simply restricting access to nutrition through a short-term limited feeding regimen (*e.g Arslan-Ergul et al., 2016*) led to decreased fish vitality and viability and was not continued. In order to avoid malnutrition, we designed unique feeds that limit the total caloric content of the food without reducing the lipid and vitamins/minerals (*Supplementary file 1A*). Observations showed that adult fish actively fed on all experimental feeds. Two separate replicate experiments were set up. In each, an equal number of young adult fish of particular genotypes were grouped into common feeding populations. In the first experiment, wild-type fish were compared to homozygous mutants (n = 17), whereas in the second experiment *frnt* siblings (i.e. +/+ and +/-) were compared (n = 30). Over the course of the experiment, there was no significant reduction in weight in the different feeding groups, however there was a marked lack of an increase in body mass in the 50% restricted feeding group (*Figure 9—figure supplement 1*).

Zebrafish fed with control feed followed the general expectation for the zebrafish lifespan, with greater than 70% survival over a 5 month period (*Figure 9A*). *Frnt* sibling controls (wild-type and heterozygous mutants) however, provided with the same feed showed a considerable shift in viability (*Figure 9B*), suggesting a potential dominant effect of *celsr1a* on long-term viability. In both experiments, 25% reduction in calories did not show any significant effect on viability in wild-type, sibling controls or *frnt* mutants (*Figure 9A,B*). However, in 50% calorie reduced groups, both homozygous *celsr1a* mutants as well as control groups showed a significant shift in lifespan (*Figure 9A,B*). As viability is a broad assessment of potential changes in aging, we looked closely at changes in phenotypes associated with loss of *celsr1a* function in mutants fed different diets. As feeding regimens were initiated in 3 month-old fish sorted by their integumentary phenotypes, scale phenotypes were found in all treated fish as they were already present at the start of feeding. Therefore, this phenotype cannot be used to assess response to caloric restriction (CR). We used behavior as an overt measure of change in aging-related degenerative phenotypes (*Figure 9C*, Suppl. *Videos 1* and *2*). We found that 50% reduced caloric intake results in a considerable reduction in the circling behavior and sharp turns observed in *celsr1a* mutants (*Figure 9D–E*), suggesting that the treatment halted or ameliorated this phenotype in the mutant. We further investigated the changes in smooth muscle and intestinal phenotypes observed in the *celsr1a* mutant in response to caloric restriction. A 50% restriction in calories led to increased slow muscle fiber area in both sibling and mutant groups (*Figure 9I*) even extending beyond adult wild-type sizes. Calorie restricted fish show a parallel shift in *sirt1* expression in the muscle (*Figure 9J*) as seen in the brain (*Figure 9F*), although non-significant due to variability across samples. 50%CR treatment also leads to significant shifts in differentiation of the enterocytes of the posterior intestine, leading to more goblet cell morphology (*Figure 9K*).

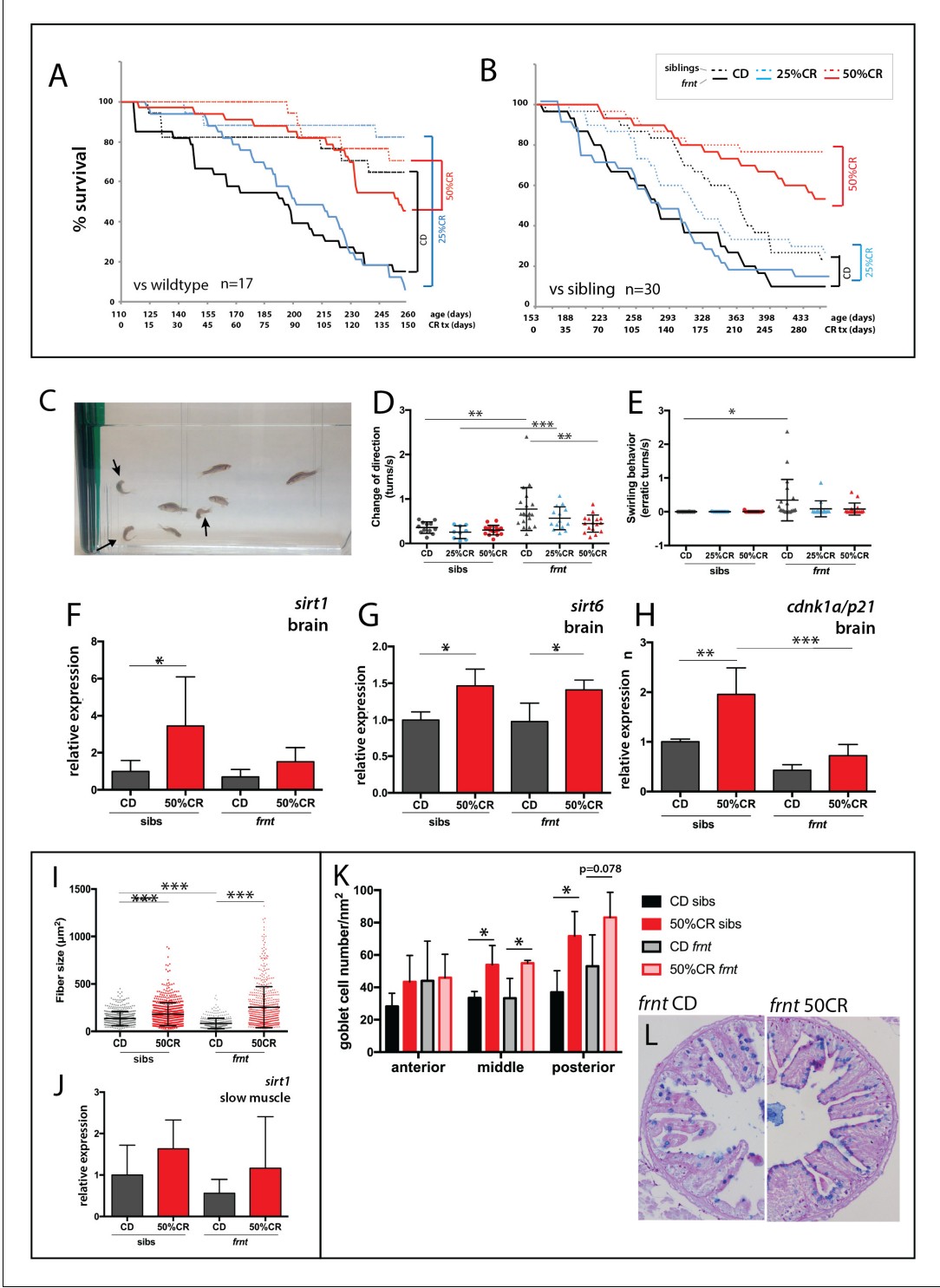

**Figure 9.** Caloric restriction increases longevity and alleviates pathology of *celsr1a/frnt* mutants. (**A–C**) Effects of specialized diets, having no restriction (control diet (CD), black), 25% (blue) or 50% (red) calorie restriction (CR), on viability of zebrafish; *solid line, frnt* mutant; *dotted line*, control fish. (**A–B**) The effect of the specific dietary reduction in calories on survival in *frnt*; 50%CR lead to significant increase in survival of mutant and siblings compared to control and 25%CR; differences between 50%CR diets for both mutant and siblings are all significant (p<0.001) compared to control and 25%CR by Mantel-Cox and Geha-Brelow-Wilcoxon tests. (**C–E**) Caloric restriction ameliorates aberrant swimming behavior in *celsr1a* mutants. (**C**) High frequency turning behavior of mutants in tank (arrows). Quantitation of change in direction (**D**) and erratic turns (**E**) in treatment groups with

*Figure 9 continued on next page*

*Figure 9 continued*

different levels of caloric reduction. (F–H) Expression of genes associated with senescence and lifespan in brain tissue of wild type and *celsr1a* mutants in different dietary treatments: (F) *sirt1*; (G) *sirt6* and (H) *cdnk1a/p21*; data represented as mean ± standard deviation. (I–J) Effect of caloric restriction on slow muscle fiber area in wild-type and *celsr1a* mutant adult zebrafish. (I) Slow muscle fiber area is potentiated in response to 50%CR in both wild-type and mutants. (J) *sirt1* expression in slow muscle from different diet treatment groups. (K–M), Effect of caloric restriction on intestinal differentiation phenotypes of wild-type *celsr1a* mutant adults. (K) goblet cell number quantitated from different regional areas of the gut in CD and 50%CR treatment groups. Data presented as mean + /- standard deviation, *p<0.05, paired t-test, n = 4 (*frnt*), n = 3 (sibs). (L) Goblet cell morphology in CD and 50% CR treated mutant posterior intestine.

The online version of this article includes the following figure supplement(s) for figure 9:

**Figure supplement 1.** Analysis of weight in caloric restricted zebrafish.
**Figure supplement 2.** Compensatory responses of *celsr1* homologues to caloric restriction.
**Figure supplement 3.** Morphology of caloric restricted zebrafish.

As previously noted, *celsr1* in the zebrafish has two paralogues, *celsr1a* and *celsr1b* as well as two orthologues, *celsr2* and *celsr3*. Intriguingly, in both siblings and homozygous *celsr1a* mutant fish, caloric restriction led to an increase in *celsr1b* expression. A significant increase in expression of *celsr2* or *celsr3* orthologues can be seen in siblings treated with 50% CR feed. Paralleling these data, in the surviving *frnt* mutants an upward trend in *celsr2* or *celsr3* gene expression is also observed (*Figure 9—figure supplement 2*). These data suggest that in tandem with an increase in metabolic regulators of aging such as *sirt1* and *cdnk1a/p21* (*Figure 9 F-H*), caloric restriction causes an upregulation in *celsr1b* even in siblings that may contribute to the observed rescue.

Although tissue integrity, behavior, and lifespan significantly improved with 50% reduction of calorie intake, overall morphology of the *frnt* mutant survivors remained generally unaffected (*Figure 9—figure supplement 3*).

## Discussion

Zebrafish have served as a highly efficient laboratory model to perform unbiased screening for the genetic regulation of embryonic and post-embryonic development. However, its use towards investigating the regulation of aging has been limited to known genetic factors identified in the mouse and modeled by reverse genetic approaches. Here, using a forward genetic approach in the zebrafish centering on phenotypes manifesting in the adult, we isolated a novel mutant class which exhibits a collection of phenotypes that together closely resemble natural aging.

In a direct comparison between *frnt* mutants with older fish showing outward appearance of senescence, we demonstrate the similarity of the mutant phenotype with normal aging pathologies in fishes. All the phenotypes noted in *frnt* are shared with other vertebrates and are seen in normal and accelerated aging in both mice and humans. Cloning of the zebrafish mutants revealed that the progressive loss of homeostasis was due to mutation in *celsr1a*, a member of the flamingo family of cadherins. Expression of *celsr1a* is found within specific tissues in developing fish and diminishes as fish mature. Thus, the loss of *celsr1a* function in the *frnt* mutant may reflect conditions occurring at later stages of adult development and homeostasis, leading to the early appearance of aging-like phenotypes.

### Conservation of *celsr1* function in vertebrates

Our identification of *celsr1a* giving rise to an adult aging phenotype in the zebrafish is surprising as loss-of-function mouse models and humans carrying mutations in *Celsr1* have a high prevalence of neural tube closure defects (*Allache et al., 2012*; *Chen et al., 2018*; *Curtin et al., 2003*; *Murdoch et al., 2014*; *Robinson et al., 2012*; *Wang et al., 2018*). As other planar cell polarity regulators are associated with neural tube defects and have been shown to genetically interact with Celsr1 to increase the severity of the pathology (*Murdoch et al., 2014*; *Wang et al., 2018*), planar cell polarity most likely plays a key role in the etiology of these disorders. We do not see neural tube defects arising in the *frnt* mutants nor do we observe reduced numbers of juvenile *celsr1a* homozygous mutants as would be expected from early lethality. Zebrafish have two paralogues of many

genes as a result of an ancestral whole genome duplication. Retention of paralogues can provide redundancy and buffering of essential functions, allowing for resolution of functions later in development (*Kassahn et al., 2009*). Although we have not specifically investigated the overlapping function of *celsr1* paralogues, such redundancy could underlie the lack of early neurulation phenotypes in the *celsr1a* mutants. Another hypothesis for the lack of neural tube deficiencies in the zebrafish *celsr1a* mutants is simply that, as teleosts form the neural tube by cavitation of the neural keel (*Papan and Campos-Ortega, 1994*) and not intercalation of neural folds as in amniotes, *celsr1a* and/or PCP is not essential for this developmental mechanism. Prior data suggested that morpholino knockdown of *celsr1a* led to neural keel defects (*Formstone and Mason, 2005*), however we do not observe these phenotypes in any of the defined *celsr1a* mutants. This early developmental difference we observe in fish may have permitted the discovery of the late developmental effects of *celsr1a* seen here and revealed a role for this gene in the regulation of aging.

Celsr1 plays several signaling roles both in planar cell polarity/non canonical Wnt, as well as, Hippo signaling. Mice with deficiencies in Celsr1 show distinct polarity defects in mouse oviduct epithelia (*Shi et al., 2014*), hair development (*Devenport and Fuchs, 2008*; *Ravni et al., 2009*) as well as patterning of tongue papillae (*Wang et al., 2016*). We see an analogous phenotype of integumentary phenotypes in the patterning and loss of asymmetry in scales of the zebrafish (*Figure 6*). Integumentary appendages, while structurally diverse, all share a common early patterning placodal stage, suggesting this may be a point at which pattering is determined by Celsr1. *Celsr1* mutants in the mouse have been found to have a dominant effect on vestibular function (*Curtin et al., 2003*). This has been shown to be associated with misoriented outer hair cell stereociliary bundles regulated by PCP signaling (*Curtin et al., 2003*). The consequence of these vestibular defects is altered stereotaxis and swirling of mouse Celsr1 mutants (e.g *crash* (*csh*) and *spincycle* (*Scy*)). We show that *celsr1a*-deficient zebrafish show comparable behavioral phenotypes with prominent circling/swirling behavior comparable to those seen in the mouse (*Figure 9—figure supplement 1*, *Video 1*). Although a detailed analysis of the vestibular system and otoliths in *frnt* mutants have not been carried out, it is likely that a similar mechanism underlies this phenotype in both species.

## Role of *celsr1a* in regulating progenitor cell populations

Although resembling normal aging, *celsr1a* mutant fish do not show significant shifts in expression of senescence biomarkers (*Figure 3*). However, many tissues show acquired deficiencies in tissue integrity and homeostasis similar to those observed in normal-aging zebrafish. These homeostatic aging phenotypes are coincident with decreased tissue specific markers of resident stem cells and proliferation. Expression analysis shows *celsr1a* diffusely expressed during early embryogenesis becoming localized to a diverse array of tissues as development progresses (*Figure 5*). Within the intestinal endoderm, *ceslr1a*<sup>GFP</sup> has heightened expression of the marker in localized basal cells. The majority of cells strongly expressing *celsr1a* co-label with *neurod1* a marker for differentiated EECs suggesting a role of these cells in the observed pathology seen in the mutants (*Figure 8A–C*). These cells are not actively cycling as they do not take up BrdU (*Figure 7J–K*). In adults the population of cells with co-expression is lower but remains a significant portion of *celsr1a*<sup>GFP+</sup> cell populations (*Figure 8*). Previous work has identified EECs as being a source of quiescent stem cells in the adult mouse intestine (*Basak et al., 2017*; *Sei et al., 2018*). EECs are sufficient to contribute to homeostatic and repair activities in the mouse in cell populations not expressing the broad stem cell factor *leucine-rich repeat-containing G-protein coupled receptor 5* (*Lgr5*) suggesting EEC may be a source of resident quiescent stem cells for this tissue. Lgr5<sup>+</sup> intestinal stem cells show a bias towards differentiation into EEC morphologies *in vitro* suggesting EECs may have developmental potential within the intestine for proliferation and stem cell function (*Basak et al., 2017*; *Buczacki et al., 2013*; *Sei et al., 2018*). Zebrafish do not have a *Lgr5* orthologue for direct comparison, however we show that *celsr1a* marks a similar population of secretory EECs and that loss of *celsr1a* function leads to a loss of homeostasis and decreased progenitor cell number. Thus, *celsr1a* may be essential for the specification of an early defined EEC population in the zebrafish comparable to those detailed in the mouse having quiescent stem cell properties (*Buczacki et al., 2013*).

In mice and zebrafish, notch signaling is required for EEC differentiation (*Flasse et al., 2013*; *Fre et al., 2005*). Inactivation of notch signaling in the mouse leads to a decrease in stem cell progenitors and overpopulation of goblet cells in the villi (*Jensen et al., 2000*; *Kokubu et al., 2008*; *Pellegrinet et al., 2011*; *Riccio et al., 2008*; *van Es et al., 2005*). Recent work has implicated a

population of non-proliferative sensory cells that are notch-responsive in the regulation of the stem cell niche during zebrafish intestinal development (*Li et al., 2019a*). In *celsr1a* mutants, we see a definitive switch of vacuolated EEC phenotypes and general differentiation of the intestinal epithelium, particularly in the posterior region (*Figure 7—figure supplement 1*). Such qualitative shift in differentiation suggests that the decrease in proliferative capacity may be due to abnormal differentiation of progenitor cells in the mutant. This shift leads from a deficiency in signaling from a *celsr1a* labeled cell populations. Although there is a mechanistic link between PCP and notch signaling (*Le Garrec and Kerszberg, 2008*), it has not been clarified if a similar relationship exists in the zebrafish intestine. Interestingly, the cells closely resemble a conserved type of vacuolated EEC recently described as Lysosome Rich Enterocytes (LREs [*Park et al., 2019*]) that are essential for nutrient uptake and transcellular transport of cargos, however specific characterization of lysosomes within *celsr1a* positive cells remains to be determined.

The role of *celsr1a* in regulation of adult stem cells may be shared in various tissues. *Celsr1* mRNA is found to be expressed in zones of neural stem cell (NSC) proliferation in the mouse and abates postnatally in parallel to decreasing numbers of NSC (*Goffinet and Tissir, 2017*). Similarly, *Celsr1* in the mouse was recently found to mark a population of quiescent mesodermal stem cells that contribute to tissue repair (*An et al., 2018*; *Sugimura et al., 2012*). Thus, the effects of *celsr1a* deficiency we observe in the intestine, skin and muscle may have broader implications to stem cell regulation in other tissues, consistent with the degenerative, acquired phenotypes we observe in the *celsr1a* mutants. We observe an association of *celsr1a* effects to highly metabolically active tissues such as the slow muscle, epidermis, and intestinal epithelium. The contrast of the pathologies observed in slow muscle (mitochondrial rich red fibers) compared to fast muscle (white fibers) in *frnt* mutants clearly states this differential. This is consistent with the spectrum of affected tissues in mouse mutants affecting telomere maintenance suggestive of a role in stem cell maintenance as a potential cause of the phenotypes observed (*Carneiro et al., 2016*; *Lee et al., 1998*). We favor hypotheses of a tissue specific role for *celsr1a* in the regulation of aging phenotypes, however given the role of EECs in hormonal regulation, it remains a possibility that the compound effects and acquired aging phenotypes observed across several tissues in the *frnt* mutant are mediated through systemic/hormonal signaling from the intestine. Such transcellular signaling of EEC cells could foster endocrine signaling that would affect regulation of proliferation and stem cell regulation in other tissues. This hypothesis for intestinal dependence of the senescent phenotypes will require further analysis of tissue specific loss-of-function of *celsr1a* and analysis of maintenance of homeostasis in adults.

## A zebrafish model of stem cell regulation and aging

Phenotypes resembling aging can be influenced by metabolism, activation/alleviation of senescent cell influences, epigenetic regulation such as methylation and acetylation, and general loss of fidelity in transcriptional regulation. Tissue homeostasis is a key factor maintaining tissue vitality and physiological function and has been proposed to be a major factor regulating age-associated phenotypes in many organs and tissues (*Liu and Rando, 2011*; *Schultz and Sinclair, 2016*). Using an unbiased genetic screen in the zebrafish, we identified a progeric mutant showing broad tissue level deterioration and loss of proliferative capacity of tissues. Our data suggest that the phenotypes are due to specific loss of progenitor cells in tissues and correlate with loss of expression of stem cell markers such as *dNp63* in the epidermis, *pax7a* in slow muscle, as well as known mammalian markers, *sox2* and *olfm4*, in the zebrafish intestinal epithelium. The identification of *celsr1a* as a causative factor underlying these phenotypes suggests that PCP signaling is essential for the appropriate maintenance of stem cells in adult tissues. We show that *celsr1a* marks EEC cells in the zebrafish, cells previously defined as having stem-like capacity in the mouse intestine. Thus, identification of *frnt* mutants having distinct progeric phenotypes reveals a new model for the regulation of maintenance of stem cells and aging in the zebrafish. It is interesting that although broadly resembling normal senescence in zebrafish, the *celsr1a* mutant phenotype exhibits a subset of core molecular and metabolic signatures of aging. This supports the notion that aging is multifactorial complex phenotype. The fact that *celsr1a* can mirror overall organismal phenotypes, also suggests that aging pathologies may be interrelated and represent a systemic readout of even tissue level-events.

It has been difficult to completely reconcile phenotypic similarities between progeria and normal processes occurring during aging (*Burtner and Kennedy, 2010*). One way to address this question

is to see if treatments thought to suppress normal aging can ameliorate the age-related pathologies observed in mutants. We designed a specific caloric-restricted diet for the zebrafish as a means to test if we could modify the *frnt* aging phenotype through modification of diet. Shifting to a restricted diet in late development showed to be quite efficacious in extending the viability of the mutant, as well as heterozygous siblings specifically in 50% caloric reduced feeds (*Figure 9*). Caloric restriction also resulted in decreased manifestation of aging phenotypes in *celsr1a* mutants including behavioral deficiencies (*Figure 9C–E*) as well as an upregulation of markers consistent with metabolic regulation of aging (*Figure 9F–H*). Interestingly, the effect of caloric restriction led to upregulation of *celsr* paralogues, suggesting that alteration of *celsr* function can compensate in part for *celsr1a* deficiencies (*Figure 9—figure supplement 2*). The reduced expressivity of the mutant phenotype by caloric restriction supports that the alterations in the *frnt* mutant cause changes that are normally modulated by pathways associated with normal aging.

## Conclusions

Through use of genetic screens in zebrafish, we have identified a novel role of *celsr1a* in stem cell function, maintenance and/or proliferation and that disruption of this regulation leads to premature aging of zebrafish. Importantly, the phenotypes detailed occur late in development and affect the early onset or expressivity of aging phenotypes. Although zebrafish are not well suited for systematic analysis of longevity due to their relatively long normal lifespan, one promising aspect of defined mutants having premature aging is their use in screens for genes or specific alleles that can specifically abrogate effects on aging or lifespan phenotypes. Such modifier screens remain a viable future research strategy and tool for discovery using this model.

# Materials and methods

## Husbandry

A complete description of the husbandry and environmental conditions in housing for the fish used in these experiments is available as a collection in protocols.io dx.doi.org/10.17504/protocols.io. mrjc54n. All experiments used both male and female fish as no obvious phenotypic difference of the mutation were noted. All experimental procedures involving fish conform to AAALAC standards and were approved by institutional IACUC committee. Mutant alleles used in this work are *celsr1a*$^{t31786}$ (R122Ins3.5kb), *celsr1a*$^{mh36}$(C1693X), *celsr1a*$^{mh104}$(P2027A-fs11X), *celsr1a*$^{GFP}$ (mh202, L74InsGFPfs), and *tert*$^{hu3430}$ (C168X). Transgenic line *Tg(neurod1:TagRFP)*$^{w69}$ was kindly provided by Dr. John Rawls.

## Fish behavior videotaping

Four tanks were placed in a 2 × 2 stack with each tank housing a single individual (Suppl *Video 2*). Single recordings were made in order to avoid the mis-tracking of individuals. Fish were put into the video tank 10 min before recording in order to allow them to acclimate. Behavior was then videotaped for 5 min periods. Behavior such as the swimming distance, velocity, time spend in the top half of the tank, change in direction (swim changes from one direction to another direction), erratic turn times (fish swirl or rapid direction changes ($\geq$2 turn/s)) in the five minutes video were recorded and analyzed by ANYMAZE (Stoelting Co.).

## Quantitative polymerase chain reaction

Tissues were isolated and immediately frozen in liquid nitrogen before storing in −80°C or put in TRI Reagent (Sigma) for RNA extraction immediately. Total RNA was extracted by TRI Reagent (Sigma) or Direct-zol RNA Miniprep Kit (Genesee Scientific), and RNA was reverse-transcribed by Superscript IV (Invitrogen) or RNA to cDNA EcoDry Premix (Oligo dT; Takara). PCR was carried out using Quanti-Fast SYBR Green PCR Kit (Qiagen) or SYBR Green PCR Master Mix (Applied Biosystems). The expression levels of target genes were normalized to the levels of reference genes, ribosomal protein L13 alpha (*rp113a*) or tubulin (*Tang et al., 2007*). The relative expression ratio of each target gene to the reference was normalized to the control group ($2^{-\Delta\Delta ct}$ method). All qRT-PCR assays in a particular experiment were undertaken at the same time under identical conditions and performed in triplicate. Primer sequences used for gene amplification are listed in *Supplementary file 1B*.

## Mutagenesis and non-complementation screen

Tuebingen male fish were treated with N-ethyl-N-nitrosourea (ENU; Sigma) following an optimized protocol using clove oil (Sigma) as a sedative (*Rohner et al., 2011*). The surviving mutagenized founders were crossed to *frnt*$^{t31786}$ homozygous females. Progeny were screened at 2–3 months of age to identify *frnt* phenotypes. Mutants were maintained by out crossing to Tuebingen wild-type strain and incrossing.

## Mapping of *frnt*

Rough mapping of the mutant *frnt* was based on a whole genome sequencing method described previously (*Bowen et al., 2012*). DNA from 20 homozygous F2 from *frnt/+* incrosses was isolated and pooled for DNA library construction. Whole-genome sequencing was carried out on an Illumina HiSeq2000, using 100 bp single-end sequencing. Linkage was confirmed and an interval was narrowed down by analysis of recombinants using microsatellites and SNP markers. To further refine candidate genes, the ENU generated allele (*mh36*) that failed to complement *frnt* was sequenced. DNA was isolated from two homozygous individuals and whole-exome sequencing was carried out using 50 bp paired-end sequencing. Three top candidates which had either missense/nonsense mutations or low coverage in the linked region in both of the alleles were chosen for CRISPR/Cas9 targeted mutagenesis. Sequencing of the non-complementing alleles *frnt*$^{t31786}$ and *frnt*$^{mh36}$ identified *celsr1a* as the likely causative gene in *frnt*$^{t31786}$. Analysis of whole genome sequencing data identified a sharp break point in the sequencing read coverage of *celsr1a* in *frnt*$^{t31786}$. To search for reads spanning this insertion, we used Blastn (*Altschul et al., 1990*) to identify reads where one half of the read had 100% match to either side of the putative insertion. We then used CAP3 (*Huang and Madan, 1999*) to perform a de novo contig assembly on the identified reads. We were unable to assemble a single contig containing the entire insert, suggesting this insert spanned a greater length than could be contained in a single 100 bp sequencing read. To identify the identity of the insert, we then performed BLAT on Ensembl against the zebrafish genome on the non-*celsr1a* portion of each contig. Through this, we discovered that the *frnt*$^{t31786}$ mutation was due to a 3.5 kb transposon insertion in exon 1 of *celsr1a* (ENSDARG00000093831). *frnt*$^{mh36}$ had a nonsense mutation in exon 8 of the same gene.

## Reverse genetic editing of *celsr1a* locus

Homozygous *frnt*$^{t31786}$ were outcrossed to Tuebingen wild-type fish and progeny (*frnt/+*) were used for complementation testing. Guide RNA (gRNA) targeting exon 16 of *celsr1a* were designed using Zifit (zifit.partners.org) (*Sander et al., 2010*). A mix of 150 ng/µl Cas9 mRNA, and 100 ng/µl gRNA was injected into *frnt/+* one-cell stage embryos in a total volume of 2 nl. Fish were screened at young juvenile stages for appearance of the *frnt* phenotype.

Green fluorescent protein (GFP) was knocked-in 114 nucleotides upstream of the start codon of *celsr1a* using CRISPR/Cas9. One gRNA targeted close to the start codon was chosen based on CHOPCHOP prediction (http://chopchop.cbu.uib.no/)(*Labun et al., 2019*). A donor plasmid was constructed using 1 kb homology arms at each side of the insertion site. A mix of 125 ng/µl Cas9 mRNA (System Biosciences), 12.5 µM gRNA (IDT), 10 ng/µl donor plasmid and 1 mM SCR7 (Xcessbio Biosciences) was injected into wild-type one-cell stage embryos in a total volume of 2 nl. Fish were screened at 24–72 hpf for the presence of GFP expression.

## Histology

Fish were anesthetized by 0.4% MS-222 and fixed using 4% paraformaldehyde (PFA) at 4°C overnight, decalcified in 14% EDTA for one week before proceeding for dehydration and embedding in paraffin. Samples were cut at a 6 µm thickness and stained with Haematoxylin (Electron Miscroscopy Sciences) and Eosin (Sigma). For Alcian blue PAS staining, after deparaffinization and hydration to distilled water, slides were stained in 1% alcian blue solution (pH 2.5) for 30 min, then washed in running tap water for 2 min and rinsed in distilled water, then were oxidized in 0.5% periodic acid solution for 5 min, rinsed in distilled water and placed in Schiff's reagent for 15 min. For PAS stain, after *in situ* hybridization slides were oxidized in 0.5% periodic acid solution for 5 min, rinsed in distilled water and then placed in Schiff's reagent for 15 min for staining.

Muscle fibers were measured from individual sections stained with Hematoxalin and Eosin using Nikon NIS Elements software package v4.4 quantitation software. At least three individuals were counted from each genotype and age group, and fibers from both the left and right slow muscle were counted. Only one section was counted for each individual. Sections of intestines were made such that all three regions of the gut can be identified on a particular section (*Figure 7—figure supplement 1A*). In this way for each experiment, analysis was centered on regional differences in the intestine and focused to comparable areas across individuals. Individuals were harvested at comparable times, but were not removed from food for extended length of time prior to euthanasia.

## *In situ* hybridization

Probes for *in situ* hybridization were synthesized using DIG RNA Labeling Kit (Roche). *In situ* hybridization was carried out on paraffin sections. Slides were rehydrated, digested by proteinase K and acetylated by treatment with acetic anhydride in triethanolamine. Sections were hybridized with approximately 10 ng probe in 100 µl hyb at 65°C overnight. After post-hybridization wash and antibody incubation with anti-Digoxigenin-AP (1:2500 dilution), the signal was detected by BCIP/NBT (Sigma).

For quantification of *olfm4* expression in intestines, we counted the average number of *olfm4* positive cells per section. Two to three sections per fish were counted; n = 5–7 fish.

## Electron microscopy

Excised samples of the flank encompassing slow muscle tissue of adult wild-type and *celsr1a* mutant fish were fixed with a mixture of 4% PFA in PBS and 1–2.5% glutaraldehyde. After post-fixation with 1% osmium tetroxide in 100 mM PBS, samples were treated with 1% aqueous uranyl acetate, dehydrated through a graded series of ethanol and embedded in Epon. Ultrathin sections were stained with uranyl acetate and lead citrate and viewed in a Philips CM10 electron microscope housed at the Max Planck for Developmental Biology, Tübingen, Germany.

## Southern blot of telomere length

Genomic DNA was extracted by phenol-chloroform-isoamyl alcohol extraction method in order to obtain intact long telomeres, and was digested by HinfI, RsaI, AluI, MspI and HaeIII. 6 ug of total digested genomic DNA was loaded per lane. After electrophoresis, DNA was transferred to a positively charged nylon membrane. Probe labeling and Southern blot detection were carried out using 'DIG High Prime DNA Labeling and Detection Starter Kit I' (Roche). Probes were generated using PCR based amplification of the (TTAGGG) repeat only were used to amply fragment from telomere and subcoloned. Probe was labeled with digoxigenin-dUTP.

## Skeletal staining and quantitation

Alizarin red staining was performed using 1% alizarin red in 0.5% KOH. Tissue was dehydrated in ethanol prior to staining. Measurement of scale diameter accomplished through quantitation tools within the Nikon NIS Elements software package v4.4.

## SA-β-gal staining

Fish tissues were fixed in 0.2% glutaraldehyde overnight, stained in 1 mg/ml 5-Bromo-4-chloro-3-indolyl β-D-galactopyranoside (X-gal, Cell Signaling) pH of 5.9–6.1 overnight at 37C, and post-fixed in 4% PFA. The samples were then processed for paraffin embedding by standard dehydration methods. Cross-sections of these samples were cut at a 6 µm thickness and counterstained with nuclear red.

## BrdU labeling

For analysis of proliferation in larvae, 10 mM BrdU (Sigma) was added to E3 buffer and larvae were treated for 24 hr at 28.5°C. For analysis in adult tissues, 30 µl of 2.5 mg/mL BrdU (Sigma) was injected intraperitoneally and samples were collected at designated times after injection (*Hui et al., 2014*; *Schall et al., 2017*).

BrdU staining was conducted following *Verduzco and Amatruda (2011)* with minor modifications. In brief, larvae were fixed in 4%PFA for 2 hr, then transferred to methanol at −20°C. A 1:100

dilution was used for anti-BrdU antibody after 5 × 10 min wash in PBST. Adult tissues were dissected, fixed overnight, and processed for standard paraffin embedding. Cross-sections of these samples were cut at a 6 μm thickness and treated with standard immunofluorescence (1:500 dilution of BrdU antibody) with an antigen retrieval step of boiling in sodium citrate buffer (10 mM, pH 6) for 5 min. Slides were then counterstained with 300 nM DAPI for 30 min. Quantitation of Brdu positive cells in cross section was normalized by number of rugae as this was found to be the most robust value among fish and is associated with areas of proliferation at the valley between rugae. Double labeling of BrdU and GFP in larvae was performed by whole mount immunological staining of *cels-r1a$^{GFP}$* for BrdU and DAPI and imaging of whole larvae intestines with confocal microscopy.

## Quantitation of co-labeling

Cell counting was automated using ImageJ 1.52 p with Fiji. All images used a gaussian blur with sigma = 1 (GFP) or sigma = 2 (RFP) followed by a rolling ball background subtraction of size 50 (GFP) or 20 (RFP) with sliding paraboloid enabled. These were used to identify maxima with prominence = 1, to be used as seeds for marker-controlled watershed. The mask for the watershed was derived using edge detection, enhanced contrast, thresholding, and the binary close, fill holes, and open. The watershed basins were used to define regions of interest, but were excluded if they were larger than 700 pixels, smaller than 55, less than 0.5 round, or less than 0.3 Circularity, or if their maximum pixel intensity was less than 10. The remaining regions of interest were overlapped between channels and binary intensity confirmed their overlap.

## Immunofluorescence

Primary antibodies and dilutions: BrdU antibody (IIB5) (Santa Cruz Biotechnology sc-32323, 1:500), Phospho-Histone H3 (Ser10) (Cell Signaling Technology 9701, 1:500), Anti-GFP rabbit IgG, Alexa Fluor 555 conjugated (Invitrogen A-31851, 1:500). Secondary antibodies and dilutions: Alexa Fluor 488 goat anti-rabbit IgG (Invitrogen A11070, 1:500), Alexa Fluor 568 goat anti-rabbit IgG (H+L) (Invitrogen A21069, 1:500), Goat anti-mouse IgG (Cy3 Abcam ab97035, 1:500), Alexa Fluor 488 goat anti-chicken IgG (Invitrogen A11039, 1:500). Before blocking, sections were boiled in sodium citrate buffer (10 mM, pH = 6) for 5 min in a pressure cooker for antigen retrieval. Slides were then counterstained with about 300 nM DAPI for 30 min.

## Caloric Restriction

Mutant or control fish were placed with an equal number of albino fish at normal rearing density (20/ liter) per each feeding group tested. Albino fish were present to serve as a balance for fish density and buffer in the case of reduced viability in the study due to the mutant genotype. Two separate experiments were carried out, one with wildtype fish and the other with siblings as controls. Fish were sorted by phenotype at juvenile stages and placed on the experimental diets starting at 5 month and 3.5 months old, respectively. The caloric restriction feed was synthesized by reducing carbohydrate and proteins but maintaining the lipid and vitamins/mineral constant to avoid malnutrition. Feed was produced in the laboratory using defined ingredients. Each diet was prepared by mixing ingredients with reverse osmosis (RO) water until a homogeneous batter was formed. The batter was then spread into a thin sheet and baked at ~212 °F until dry (~1.5 hr) in a commercial convection oven. The diet was then cooled to room temperature, crushed into pieces and ground into crumble form using a bur grinder. Crumbles were then sieved into two size ranges: 100–200 micron for juvenile and 200–300 micron for adult life stages. The ingredient and constitution of the feed is shown in **Suppl. File 1A**. To normalize feeding regimes, the amount of food was provided as a measure of total weight of the fish. Fish weight was measured *en masse* per group and not singularly every two weeks, and fish were fed at 3% of total fish weight. Fish numbers were counted weekly and any deaths were recorded daily. Experiments were terminated once the percentage of fish remaining dropped below 20% of starting numbers.

## Statistical analysis

Values are shown as mean ± standard deviation. Statistical significance between two groups was determined by student's t-test. Statistical significance among several experimental groups was determined by one-way analysis of variance (ANOVA). Significance was set at $p < 0.05$. Mantel-Cox

and Geha-Brelow-Wilcoxon tests were used to plot viability curves. To compare swimming behavior data, the two-way ANOVA was used. All statistics were executed using Prism software package. Significance was set at $p < 0.05$.

## Acknowledgements

Work was supported by Ellison Medical Foundation, and Glenn Foundation awards to MPH and partially supported by grant NIH 2R01DE019837-09 (JTS/MPH). The authors wish to thank expert help of Dr. Heinz Schwartz and Iris Koch (Max Planck for Developmental Biology) for electron microscopy assistance and Ines Gehring for assistance in early positional mapping of the *frnt* mutant.

## Additional information

### Funding

| Funder | Grant reference number | Author |
|---|---|---|
| Ellison Medical Foundation | | Matthew P Harris |
| Glenn Foundation for Medical Research | | Matthew P Harris |
| National Institutes of Health | 2R01DE019837-09 | Matthew P Harris |

The funders had no role in study design, data collection and interpretation, or the decision to submit the work for publication.

### Author contributions

Chunmei Li, Conceptualization, Resources, Data curation, Formal analysis, Validation, Investigation, Visualization, Methodology; Carrie Barton, Robyn L Tanguay, Resources, Methodology; Katrin Henke, Data curation, Formal analysis, Validation, Investigation, Visualization, Methodology; Jake Daane, Data curation, Formal analysis; Stephen Treaster, Data curation, Software, Formal analysis, Methodology, Writing - review and editing; Joana Caetano-Lopes, Data curation, Formal analysis, Investigation, Visualization, Methodology; Matthew P Harris, Conceptualization, Resources, Data curation, Formal analysis, Supervision, Funding acquisition, Validation, Investigation, Visualization, Methodology, Project administration

### Author ORCIDs

Katrin Henke  http://orcid.org/0000-0002-1282-3616
Joana Caetano-Lopes  https://orcid.org/0000-0003-0310-4641
Matthew P Harris  https://orcid.org/0000-0002-7201-4693

### Ethics

Animal experimentation: This study was performed in strict accordance with the recommendations in the Guide for the Care and Use of Laboratory Animals of the National Institutes of Health. All of the animals were handled according to approved institutional animal care and use committee (IACUC) protocols of Boston Children's Hospital #3215.

### Decision letter and Author response

Decision letter https://doi.org/10.7554/eLife.50523.sa1
Author response https://doi.org/10.7554/eLife.50523.sa2

## Additional files

### Supplementary files

• Supplementary file 1. Supplmentary tables. (**A**) Defined diets for zebrafish dietary restriction. (**B**) Primers used in the study.

- Transparent reporting form

## Data availability

All data generated or analyzed during this study are included in the manuscript and supporting files.

---

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
