## [Decision Letter]

**Acceptance summary:**

Your manuscript demonstrates how forward genetic approaches in the zebrafish can be applied to discover genetic regulators of aging. While prior work has looked at larval stress as a proxy, this is the first use of zebrafish genetics to screen for factors controlling aging. This innovative approach reveals a surprising role for *celsr1a*, a non-classical cadherin associated with planar cell polarity, in stem/progenitor cell maintenance. This is supported by premature aging phenotypes in muscle, intestine, skin, and scales, as well as behavioral deficits that emerge during young adult stages in *celsr1a* mutants. These data provide support for hypotheses that stem cell dynamics control the onset and expressivity of aging. Overall, this mutant phenotype is quite interesting, in particular the ability of caloric restricted diets to partially rescue the phenotype. Such dietary modulation of aging phenotypes is a first for zebrafish. This work will be of interest to readers in the fields of developmental biology, stem cell biology, and aging, and provides a framework for cross-disciplinary analysis of aging and its regulation.

**Decision letter after peer review:**

Thank you for sending your article entitled "Celsr1a is essential for tissue homeostasis and onset of aging phenotypes in the zebrafish" for peer review at *eLife*. Your article has been evaluated by three peer reviewers, one of whom is a member of our Board of Reviewing Editors, and the evaluation has been overseen by Didier Stainier as the Senior Editor.

All reviewers appreciate the innovation of this work identifying new and very interesting aging-related phenotypes in this Cels1ra mutant, but have concerns that the analysis remains fragmented in terms of delving deeply into any particular mechanism in any particular tissue. The reviewers have identified several major concerns that could be addressed to more fully validate some of the major conclusions.

Reviewer #1:

This is an interesting and well-written paper describing a detailed phenotypic characterization of the a novel premature aging mutant in zebrafish. This is the first such mutant to arise from forward genetic screens in zebrafish. The manuscript includes analysis of a number of interesting phenotypes in these mutants including muscle, intestine, skin, scale, and behavioral deficits that emerge during young adult stages. Overall, this mutant phenotype is quite interesting, in particular the ability of the caloric restriction treatment to partially rescue the phenotype, and a useful contribution to the ageing field. However, the complex multi-tissue phenotypic analysis of these mutants results in a fairly fragmented picture of the associated pathology that develops in this mutant. The reader is left unsure which tissues and cells actually require the *ceslr1a* gene to promote normal ageing (as expression of *ceslr1a* is quite widespread), though it's clear that progenitor/stem populations in the above tissues are affected directly or indirectly. Below I list several concerns and potential approaches to improve the clarity and impact of this innovative work.

1) Figure 7—figure supplement 1E-H and associated text: Please clarify what region of the intestine was used to generate these images. These single images are not sufficient to support the claim in the first paragraph of the subsection “*celsr1a* is required for proliferative capacity and maintenance of intestinal progenitor cells” that there are more goblet cells in the mutant. That would need to be quantified across multiple animals, with clarity on what intestinal regions were assessed. Indeed, when I look at other rugae in panels E and G, it looks like mutants may have *less* goblet cells, so this definitely needs to be more rigorously quantified if the data is to stay in this paper. Also of potential interest, it appears (again, from just these single images) that the mutant intestinal epithelium has aberrant structure beyond goblet cells. To me it looks like there may be an enrichment of lysosome-rich enterocytes (LREs) which were very recently described by Michel Bagnat's group (Park et al., 2019). If the authors are indeed able to bring in quantitation from other animals to look at goblet cell number, please also keep an eye on those LRE-like cells to see if that is consistent too. When these edits are made, please also adjust the Discussion text reviewing these data.

2) Subsection “Role of *celsr1a* in regulating progenitor cell populations”, first paragraph: Figure 7K shows colocalization of *neurod1^+^* gut epithelial cells (presumably enteroendocrine cells) with *ceslr1a^GFP^* at 4dpf – this alone is not enough to posit here that enteroendocrine cells are contributing to a gut pathology that doesn't emerge until months later. I think the authors would need to evaluate that colocalization at the later adult stages in order to sufficiently strengthen that claim. It would also be helpful if the authors could quantify this colocalization rather than just providing a single image. Similarly, the lack of BrdU incorporation at this early 4dpf stage doesn't have strong relevance for how those cells might behave in adult stages. So, again, I think this would need to be repeated in (young?) adult intestine. As mentioned elsewhere in this review, this recent paper from Ken Wallace's lab (Li et al., 2019a) might be relevant here, as it reports a population of non-proliferating secretory cells in the zebrafish larval intestine that might regulate development of the intestinal stem cell niche. Could these be the same *neurod1^+^/celsr1a^+^* cells reported here?

3) Figure 8E, F: The behavioral phenotype in control diet mutants appears to be driven by just 1-2 animals, with no similar outliers appear in the 25% or 50% CR diet mutants. This seems like a borderline phenotype that may not be worth reporting here. What about some of the other phenotypes reported earlier in the paper, in the muscle and intestine for example? Are they rescued by the 50% CR? As a side note, the behavioral phenotype is interesting, but the absence of any pathological insight there leaves it dangling awkwardly in the narrative. Are there any CNS or vestibular defects apparent by histology or TEM?

4) Figure 9: The authors claim in the text that this shows compensatory upregulation of *celsr* homologs in *celsr1a* mutants, but none of these paralogs show statistically significant upregulation by 50% CR in the mutant (whereas they are all robustly induced in WT animals). Therefore there's certainly no sign of "compensation" in the mutant, and the overall claim that *celsr* homologs might be functionally compensating for *celsr1a* mutation seems inadequately supported here. I recommend removing these data from the paper, or at least relegating them to the supplementary section and a more minor point in the text. Please also remove/adjust the associated claim in the Discussion on this point.

5) Considering the fragmented/multi-tissue nature of the mutant phenotypic analysis reported in the earlier part of the paper, the inclusion of the caloric restriction experiment at the end was quite exciting, as it seems to provide an opportunity to see how each affected tissue responds, potentially helping to prioritize future studies. However, the CR fish were only assessed for behavioral and gross anatomical phenotypes. I agree that scale phenotypes may take a while to recover, but why not also look at intestinal or muscle histology (which were deeply phenotyped earlier in the paper under normal feeding conditions)? Also, if you have a sense of whether there are vestibular defects, those could be inspected more thoroughly in CR too. Some sort of further phenotypic assessment of this CR rescue would significantly increase the impact of this article.

6) Figure 5: The authors' argument that these data support that *celsr1a* expression is reduced during aging, perhaps lingering in select cells, is not sufficiently supported by the data here. The qPCR data shown in panel A show that all tissues have apparent reduction in *celsr1a* expression by 9 months of age compared to earlier expression, but this appears to be all relative to a given housekeeping gene (which one? Two are listed in Materials and methods). I think this would need to be confirmed with a second housekeeping gene. Further, the image data here don't convey reduced expression in multiple tissues – young and old animals' GFP levels look rather similar, actually. Can GFP levels be quantified (by qPCR and/or imaging) at different ages in select tissues? Also, it's unclear if these data are from a single or multiple animals – this should be clarified in the legend. I think these data should be improved in these ways, or the associated claim will need to be removed (and I do NOT think that would significantly detract from the paper).

Reviewer #2:

In this study, Li et al. identify *celsr1a* as the causative gene underlying mutant defects in skin and muscle homeostasis reminiscent of phenotypes that would be expected in progeric zebrafish. The authors find reduced survivorship of mutants and also defects at histological, cellular, and molecular levels in skin, slow muscle and gut. They clone the affected gene, verifying its identity by a combination of mapping, non-complementation screening, and targeted mutagenesis, and they generate an insertional GFP line to examine expression. Li et al. further provide evidence for defects in stem cell maintenance and cycling, identify a scale phenotype implicating *celsr1a* in PCP pathway signaling in zebrafish, and find that a caloric restriction diet partially ameliorates the viability phenotype, as well as behavioral phenotypes, though not the overt morphological defects. Finally the authors detect an upregulation of senescence associated genes as well as *celsr* homologues in the context of caloric restriction.

This is an interesting study that identifies zebrafish as a tractable model for studies of aging phenotypes. In addition to the analyses presented, the work raises, but does not answer, a variety of interesting mechanistic questions (e.g., how precisely *celsr1a* mutation leads to mitochondrial or other cellular defects; how caloric restriction specifically leads to upregulation of homologues and other aging genes, and whether upregulation of these genes indeed explains the partial rescue of survivorship). These and similar sorts of questions would be nice to answer but, given the time frame involved, also seem to be outside the reasonable scope of this study.

Reviewer #3:

In "Celsr1a is essential for tissue homeostasis and onset of aging phenotypes in the zebrafish" by Li and colleagues, the authors undertake a genetic screen to identify new mutants exhibiting features of premature aging and identify such a phenotype caused by mutations in the gene *celsr1a*. The overall goal of the paper is important in both identifying potentially novel pathways involved in aging and stem/progenitor cell maintenance in adult tissues as well as in exploring the utility of the vertebrate zebrafish model in characterizing the underlying molecular mechanisms. While the text and figures are extensive, the depth of the analysis on each major aspect of the mutant phenotype is, generally, either preliminary or has important limitations in interpretation due to experimental issues.

Overall, the manuscript would seem better served to focus on fewer phenotypes/related pathways and investigate these in greater depth and with increased rigor. e.g. In the Results sections, the PCP phenotype almost seems out of place initially as the paper is set up to discuss aging and stem cell maintenance, and then has this extensive section on epidermal appendages (all nicely presented, but lacking context). The Discussion then does have a lengthy section on PCP pathway in the stem cell context, but the current structure of the paper is confusing and new mechanistic insights that can be drawn are limited with the current data. Other major points include:

1) How are the authors normalizing expression data to show decreased *ΔNp63* and *pax7a*? Agreed skin is thinner and slow muscle is thinner. Doing bulk qPCR on these tissues could be misleading since the total fraction of cells of each type is not the same in each sample. The in situ staining in the gut for putative stem cell markers *Sox2* and *olfm4* seems a better assessment for overall decreased numbers of cells expressing these markers. Also, another control should be 2.5-year-old WT skin that, as shown in Figure 2E, is thinner than WT at 1 year. What happens to *ΔNp63* expression levels? Evaluating *ΔNp63* expression levels by qPCR in aged WT skin (which is also thin) would sufficiently address/aid in the interpretation of the *ΔNp63* qPCR levels of the young WT and mutant skin.

2) Most likely, *celsr1a* protein is not being made, but the Western blot in Figure 4H lacks loading controls and makes interpretation difficult. Further, what is the predicted size of the protein, and how does it match with the multiple bands identified? Why is the major band in *frnt* Mc lane higher than the major band in wt Mc? Is this a cross-reactive protein in the Mc *frnt* sample, which is expected to not make viable *celsr1a* protein? I would support that the Western needs a loading control, and Materials and methods section needed describing/showing validation of the antibody used, if the Western is ultimately included.

3) In principle, the knock-in of GFP to the *celsr1a* locus is an elegant way to monitor its expression, but the statement "The identified line, *celsr1a^GFP^*, recapitulates early expression seen by whole mount in situ (1dpf, Figure 5C)" requires a supporting in situ to the endogenous transcript in this line. I think to claim that the GFP knock-in reporter recapitulates the endogenous *cels1ra* expression pattern, an in situ to endogenous *cels1ra* (in situ probe published) is needed compared to reporter GFP (fine if separate sib embryo done by fluorescent microscopy, no need for GFP double in situ).

4) The use of label retention as a functional read-out for stem/progenitor cells in the gut is interesting, but why are Anterior BrdU cells/rugae increased at 48 hours of chase in *frnt* mutants relative to earlier time points? Even with decreased proliferation in the *frnt* mutants, should not all label-retaining populations stay constant or decreased over time? There may be variability in the degree of baseline labeling for different fish/replicates, but otherwise seems pretty consistent in other regions of the gut and at other time points in other conditions. A comment is sufficient in the Results/Discussion explaining the changes in label retention over time between the earliest and 48 hour time point for the mutant group. How is there increase in label retaining cells over time from 5 to 48 hours in the anterior segment in the *cels1ra* group? Technical issue? Perhaps different regions were assessed at the different time point?

[Editors' note: further revisions were suggested prior to acceptance, as described below.]

Thank you for resubmitting your work entitled "Celsr1a is essential for tissue homeostasis and onset of aging phenotypes in the zebrafish" for further consideration by *eLife*. Your revised article has been evaluated by Didier Stainier as the Senior Editor, and a Reviewing Editor.

The manuscript has been improved but there are some remaining issues that need to be addressed before acceptance, as outlined below:

In response to reviewer 1's concern 1, you added new data on intestinal goblet cell number (Figure 9K, L). The new data shows that the *frnt* genotype does not alter the observed increase in goblet cell number that occurs with CR. This is useful new data, but there appears to be no statistical analysis to support the claim that CR increases goblet cell number (for either genotype or intestinal region). Please add analysis of statistical significance to support this claim, or state in the legend that those differences did not reach statistical significance. Also, there is no panel M in this figure anymore, so please edit the figure legend accordingly.

---

## [Author Response]

Reviewer #1:This is an interesting and well-written paper describing a detailed phenotypic characterization of the a novel premature aging mutant in zebrafish. This is the first such mutant to arise from forward genetic screens in zebrafish. The manuscript includes analysis of a number of interesting phenotypes in these mutants including muscle, intestine, skin, scale, and behavioral deficits that emerge during young adult stages. Overall, this mutant phenotype is quite interesting, in particular the ability of the caloric restriction treatment to partially rescue the phenotype, and a useful contribution to the ageing field. However, the complex multi-tissue phenotypic analysis of these mutants results in a fairly fragmented picture of the associated pathology that develops in this mutant. The reader is left unsure which tissues and cells actually require the ceslr1a gene to promote normal ageing (as expression of ceslr1a is quite widespread), though it's clear that progenitor/stem populations in the above tissues are affected directly or indirectly. Below I list several concerns and potential approaches to improve the clarity and impact of this innovative work.

We thank the reviewer for the interest in our findings and their insight in to how to improve the data. In response to the discussion concerning scope of the characterization, I would refer to the response above concerning our decision on why we chose a broader description of aging phenotypes in the mutant. We agree a deep-dive on a particular tissue would be insightful, but we first have to show plausible mechanism of aging. Once the broad phenotype is presented and accepted by the field as resembling aging, individual organ systems can be studied in depth for effect of loss of *celsr1a* in local context.

In this work, we were successful in identifying the maintenance of stem cells and markers as key factor that is shared among tissues as a potential cause of the aging phenotype. We have not done experiments to test loss of *celsr1a* only in one tissue (e.g. loss in endothelium of gut only) to assess the essential function of that tissue to maintain normal (non-aging) phenotype. Beyond the technical complexity of this approach, it is not clear if there will be a central cause or additive causes that contribute to aging phenotypes.

1) Figure 7—figure supplement 1E-H and associated text: Please clarify what region of the intestine was used to generate these images.

We have added position in this supplementary figure as well as throughout text.

These single images are not sufficient to support the claim in the first paragraph of the subsection “celsr1a is required for proliferative capacity and maintenance of intestinal progenitor cells” that there are more goblet cells in the mutant. That would need to be quantified across multiple animals, with clarity on what intestinal regions were assessed. Indeed, when I look at other rugae in panels E and G, it looks like mutants may have less goblet cells, so this definitely needs to be more rigorously quantified if the data is to stay in this paper.

We fully agree with the reviewer statement. We have quantitated this phenotype (now added to Figure 9) and the initial observation of increased numbers of goblet cells was in fact not specific rather a general trend. However, in work provided in review, we find that alteration by 50% CR caused regional increase in goblet cell number suggesting a shift in intestinal response in the treatment. As a general description, the vacuolated cell phenotype is qualitatively distinct, just very difficult to quantify without different specific staining methods. We have changed the text to reflect these new data and implications.

Also of potential interest, it appears (again, from just these single images) that the mutant intestinal epithelium has aberrant structure beyond goblet cells. To me it looks like there may be an enrichment of lysosome-rich enterocytes (LREs) which were very recently described by Michel Bagnat's group (Park et al., 2019). If the authors are indeed able to bring in quantitation from other animals to look at goblet cell number, please also keep an eye on those LRE-like cells to see if that is consistent too. When these edits are made, please also adjust the Discussion text reviewing these data.

We agree entirely. The *celsr1a* mutant causes major shifts in posterior intestinal vacuolated cell populations as noted by the reviewer. The pictures while of a single individual are representative of the overall phenotype of the mutant. We were not able to devise a robust means to quantitate these phenotypes. We looked at goblet cell # (now in Figure 9) both in the mutant and in CR treated fish. But the boundaries of vacuolated cells were hard to pin down with histology alone. We had not seen the work on LREs prior to submitting our manuscript. We have gone back to add appropriate citation of this work in our description of the phenotype. We have also noted the transcellular signaling function of these cells in potential models of regulation of broader stem cell function in other tissues.

2) Subsection “Role of celsr1a in regulating progenitor cell populations”, first paragraph: Figure 7K shows colocalization of neurod1^+^ gut epithelial cells (presumably enteroendocrine cells) with ceslr1a^GFP^ at 4dpf – this alone is not enough to posit here that enteroendocrine cells are contributing to a gut pathology that doesn't emerge until months later. I think the authors would need to evaluate that colocalization at the later adult stages in order to sufficiently strengthen that claim. It would also be helpful if the authors could quantify this colocalization rather than just providing a single image.

We have confirmed that we can see colocalization of *neurod1* and *celsr1a:GFP* in adult intestinal epithelia. We have added both pictures and quantitation of colocalization in adults in the revised manuscript.

Similarly, the lack of BrdU incorporation at this early 4dpf stage doesn't have strong relevance for how those cells might behave in adult stages. So, again, I think this would need to be repeated in (young?) adult intestine.

We repeated BrdU incorporation in adult *celsr1a:GFP* fish to look at the retention in the intestine and its association with Celsr1 expression. Unfortunately, due to increased autofluorescence in the fixed adult tissue, the results were inconclusive and could not be added to the manuscript. We appreciate the reviewers statement that early developmental associations do not necessarily reflect those occurring in later stages and or in homeostasis. We maintain that the result in the larvae are meaningful, however we have qualified the description in the text to reflect the early developmental stage of these observations. It now reads:

“BrdU incorporation in intestinal epithelium of larvae in which *celsr1a* cells are marked with GFP (*celsr1a^GFP^*) shows restricted incorporation of BrdU in *celsr1a^+^* cells during growth (Figure 7J). This suggests that *celsr1a-*expressing cells in the larval intestine are not actively cycling”

We do not think this passage overstates the findings.

As mentioned elsewhere in this review, this recent paper from Ken Wallace's lab (Li et al., 2019a) might be relevant here, as it reports a population of non-proliferating secretory cells in the zebrafish larval intestine that might regulate development of the intestinal stem cell niche. Could these be the same neurod1^+^/celsr1a^+^ cells reported here?

The papers from the Wallace lab that were published in the last few months are very interesting. The antibody as described in the manuscript is broad in its detection of different sensory types, such that many EEC cell types all are antibody 2F11 positive. Further selection showed nkx2.2 expression further subdivides these lineages. We were unable to fully define the overlap of *celsr1a^+^* cells with the notch responsive lineage. This would be best done with the conditional transgene described in Li et al., 2019. It is a very provocative hypothesis, however beyond the *neurod1:celsr1a* overlap, which is partial, we were unable to dissociate these cells further. Single cell data of zebrafish intestinal cells would be intriguing, however, in cursory analysis of datasets (such as Lickwar et al., 2017) showed *celsr1a* to be lowly expressed, and therefore underpowered to partition into differential EEC subtypes. FACS sorted *celsr1^+^* intestines and bulk mRNAseq could approach this question in future work. Importantly, we provide the models, transgenic line, and context in which to follow these questions.

3) Figure 8E, F: The behavioral phenotype in control diet mutants appears to be driven by just 1-2 animals, with no similar outliers appear in the 25% or 50% CR diet mutants. This seems like a borderline phenotype that may not be worth reporting here. What about some of the other phenotypes reported earlier in the paper, in the muscle and intestine for example? Are they rescued by the 50% CR? As a side note, the behavioral phenotype is interesting, but the absence of any pathological insight there leaves it dangling awkwardly in the narrative. Are there any CNS or vestibular defects apparent by histology or TEM?

We thank the reviewer for their comment and agree that further investigation into the other tissues was warranted. We now include analysis of effect of CR on slow muscle and intestinal phenotypes observed in the *celsr1a* mutant and have integrated these new data with the qPCR analyses of the same tissue. The results clearly show restoration of the sarcopenia phenotype in both siblings and mutants by 50% CR treatment. Additionally, we show distinct shift in regional differentiation of goblet cells (AB+) in the intestine. These data are now included in a new Figure 9 in the revision.

The behavioral difference is actually quite robust driven by broad shift in the mean of the whole population in case of ‘turns per second’ and reduction by most of the broader “swirling” behaviors observed in the mutant. Behavior is one of the primary outward phenotypes of the fish (a consistent means of screening adults is to tap on the tank) and thus we feel it is important to include this in the CR analysis. This is nicely paired with specific changes in expression of known metabolic regulators associated with aging in other vertebrates in brain tissue of 50% CR treated and control fish.

We have previously looked for CNS and vestibular defects in the *celsr1a* mutant fish through our own analyses as well as sharing the mutants with laboratories that specialize in inner ear and neural function. No obvious pathologies were noted. TEM may reveal more nuanced age-dependent pathologies associated with inner ear, however the extensive work required to test the presence of subtle cilial defects in the mutant and their progressive deterioration is a substantial undertaking and outside the realm of this initial description.

4) Figure 9: The authors claim in the text that this shows compensatory upregulation of celsr homologs in celsr1a mutants, but none of these paralogs show statistically significant upregulation by 50% CR in the mutant (whereas they are all robustly induced in WT animals). Therefore there's certainly no sign of "compensation" in the mutant, and the overall claim that celsr homologs might be functionally compensating for celsr1a mutation seems inadequately supported here. I recommend removing these data from the paper, or at least relegating them to the supplementary section and a more minor point in the text. Please also remove/adjust the associated claim in the Discussion on this point.

We appreciate the reviewer’s comment on the measure of change seen in expression of *celsr* paralogues in the CR treated fish. We agree that the effect was modest, but the fold change was consistent between siblings and mutants for each gene. For cases that fell ‘below’ significance, the value was p<0.057. It is hard to argue that the 0.007 difference is enough to warrant removal of the data. In response to reviewers concern, we have moved the data to a supplementary figure (Figure 9—figure supplement 3) and have worked to shape conclusions present in the manuscript including choice of wording apart from ‘compensation’ as this seems to have other baggage these days.

5) Considering the fragmented/multi-tissue nature of the mutant phenotypic analysis reported in the earlier part of the paper, the inclusion of the caloric restriction experiment at the end was quite exciting, as it seems to provide an opportunity to see how each affected tissue responds, potentially helping to prioritize future studies. However, the CR fish were only assessed for behavioral and gross anatomical phenotypes. I agree that scale phenotypes may take a while to recover, but why not also look at intestinal or muscle histology (which were deeply phenotyped earlier in the paper under normal feeding conditions)?

As stated in response to question #3 above, we have added more data on slow muscle phenotypes in the CR treated fish. We’ve added slow muscle/sarcopenia and intestinal differentiation effects to CR treatment which support rescue/compensational ability of 50% CR on age-related pathologies of the *celsr1a* fish. As this paper is the first to use CR on zebrafish (not dietary restriction) we remain excited about the level of characterization we provide and the evidence of rescue/abrogation of the aging phenotypes in the *celsr1a* mutant fish.

Also, if you have a sense of whether there are vestibular defects, those could be inspected more thoroughly in CR too. Some sort of further phenotypic assessment of this CR rescue would significantly increase the impact of this article.

We are not aware of a vestibular phenotype in the mutant involving gross anatomical structure. Cilia phenotypes unfortunately would require new samples as fixation is a problem with the samples we have. Such samples would require greater than a year to procure. In response to our review plan, it was stated that this was not necessary for this particular review. We have established a broader long-term collaboration to look at this phenotype.

6) Figure 5: The authors' argument that these data support that celsr1a expression is reduced during aging, perhaps lingering in select cells, is not sufficiently supported by the data here. The qPCR data shown in panel A show that all tissues have apparent reduction in celsr1a expression by 9 months of age compared to earlier expression, but this appears to be all relative to a given housekeeping gene (which one? Two are listed in Materials and methods). I think this would need to be confirmed with a second housekeeping gene. Further, the image data here don't convey reduced expression in multiple tissues – young and old animals' GFP levels look rather similar, actually. Can GFP levels be quantified (by qPCR and/or imaging) at different ages in select tissues? Also, it's unclear if these data are from a single or multiple animals – this should be clarified in the legend. I think these data should be improved in these ways, or the associated claim will need to be removed (and I do NOT think that would significantly detract from the paper).

To compare expression levels and robustness to normalization methods, we analyzed *celsr1a* expression in young (1dpf) and aged (1 year old) adult tissue and compared it to two different housekeeping genes (Author response image 1). There was no significant difference among housekeeping gene used in analysis. In response to comments here, we have deleted the qPCR developmental analysis as we agree that it was not absolutely necessary.

**Author response image 1. respfig1:** Expression of housekeeping genes in late development and aging.

We have added photos of skin, muscle, and intestine of adult fish to Figure 5 to more clearly show decreased expression in adult tissues, then supported by the qPCR results we present. The samples are all mixed pools from several individuals. These numbers are detailed in the figure legend.

Reviewer #2:In this study, Li et al. identify celsr1a as the causative gene underlying mutant defects in skin and muscle homeostasis reminiscent of phenotypes that would be expected in progeric zebrafish. The authors find reduced survivorship of mutants and also defects at histological, cellular, and molecular levels in skin, slow muscle and gut. They clone the affected gene, verifying its identity by a combination of mapping, non-complementation screening, and targeted mutagenesis, and they generate an insertional GFP line to examine expression. Li et al. further provide evidence for defects in stem cell maintenance and cycling, identify a scale phenotype implicating celsr1a in PCP pathway signaling in zebrafish, and find that a caloric restriction diet partially ameliorates the viability phenotype, as well as behavioral phenotypes, though not the overt morphological defects. Finally the authors detect an upregulation of senescence associated genes as well as celsr homologues in the context of caloric restriction.This is an interesting study that identifies zebrafish as a tractable model for studies of aging phenotypes. In addition to the analyses presented, the work raises, but does not answer, a variety of interesting mechanistic questions (e.g., how precisely celsr1a mutation leads to mitochondrial or other cellular defects; how caloric restriction specifically leads to upregulation of homologues and other aging genes, and whether upregulation of these genes indeed explains the partial rescue of survivorship). These and similar sorts of questions would be nice to answer but, given the time frame involved, also seem to be outside the reasonable scope of this study.

The concerns about mechanism are on point. As stated in the general response to review (above) this is the nature of establishing a broad phenotype such as aging which affects many tissues. The aging field has spent decades trying to pin point a mechanistic regulation of even the ‘big hitter’ genes such as Sirt or FoxO with new mechanisms appearing almost daily. We go to extended lengths to parse out early versus later effects in manifestation of the aging phenotype of a novel zebrafish model to look at resident stem cell maintenance. These findings, through characterization of this new mutant, provide the foundations for extended analysis on detailed cell behavior regulating aging of these tissues.

Reviewer #3:In "Celsr1a is essential for tissue homeostasis and onset of aging phenotypes in the zebrafish" by Li and colleagues, the authors undertake a genetic screen to identify new mutants exhibiting features of premature aging and identify such a phenotype caused by mutations in the gene celsr1a. The overall goal of the paper is important in both identifying potentially novel pathways involved in aging and stem/progenitor cell maintenance in adult tissues as well as in exploring the utility of the vertebrate zebrafish model in characterizing the underlying molecular mechanisms. While the text and figures are extensive, the depth of the analysis on each major aspect of the mutant phenotype is, generally, either preliminary or has important limitations in interpretation due to experimental issues.Overall, the manuscript would seem better served to focus on fewer phenotypes/related pathways and investigate these in greater depth and with increased rigor. e.g. In the Results sections, the PCP phenotype almost seems out of place initially as the paper is set up to discuss aging and stem cell maintenance, and then has this extensive section on epidermal appendages (all nicely presented, but lacking context). The Discussion then does have a lengthy section on PCP pathway in the stem cell context, but the current structure of the paper is confusing and new mechanistic insights that can be drawn are limited with the current data.

We thank the reviewer for the thoughtful consideration of the papers strengths and potential weaknesses. We do appreciate the need to ‘dive-deep’ in mechanisms. As addressed in the more global response above, this paper was to define the novelty of the use of forward genetics to address aging in zebrafish – something not shown previously. By necessity, the manuscript was framed to address if this mutant is reflecting aging phenotypes, and if so, general models of aging that it might be working through. As such, we touched on a wide array of metabolically active tissues and the effect on their maintenance. Of these, the intestine was best suited for analysis of cell type and stem cell populations. But as stated by reviewer 1, even there the markers and prior evidence from the zebrafish gut are not as robust as in the mouse. Our data pushed quite far given these limitations, but this was not intended to be a paper on intestine and the specifics of signaling between different cell populations in development, homeostasis and aging. Further analyses on aging in particular tissues are currently ongoing not only in my lab but several others in the community to assess tissue level detail with specialized tools that these labs have generated. It is precisely the aim of the paper and research agenda overall to establish a zebrafish aging model that can foster such future work. I think we have done well in our design of the approach and present extremely novel data both for the aging field and zebrafish modeling of clinically important phenotypes. Thus, while we feel that the diversity of characterization is essential, in response to the reviewer’s comments we have worked to providing contextual statements as to why.

Other major points include:1) How are the authors normalizing expression data to show decreased ΔNp63 and pax7a? Agreed skin is thinner and slow muscle is thinner. Doing bulk qPCR on these tissues could be misleading since the total fraction of cells of each type is not the same in each sample.

First question: we have normalized expression data using two housekeeping genes which show consistent expression through embryogenesis and late stages of development (see Author response image 1). Following as a response to second comment, as even thin epidermis or muscle tissue will have stem cells, we are using qPCR of cell specific markers as support of histological data showing fewer stem cells.

The in situ staining in the gut for putative stem cell markers Sox2 and olfm4 seems a better assessment for overall decreased numbers of cells expressing these markers.

We agree with the reviewer that pairing this with analysis of spatial gene expression provides very clean assessment.

Also, another control should be 2.5-year-old WT skin that, as shown in Figure 2E, is thinner than WT at 1 year. What happens to ΔNp63 expression levels? Evaluating ΔNp63 expression levels by qPCR in aged WT skin (which is also thin) would sufficiently address/aid in the interpretation of the ΔNp63 qPCR levels of the young WT and mutant skin.

We have done this experiment, see Author response image 2. The results are comparable to the levels seen in the mutant supporting our argument. The histology is the primary evidence supporting the loss of stem cells and pathology. We follow with the qPCR as supporting evidence. While we agree the p63 data is consistent with the argument that the mutant resembles old phenotypes, we did not have enough samples to perform the same for each *pax7, p21* and *claudin b*. Thus, we feel that inclusion of these data without the rest will cause difficulties in the narrative.

**Author response image 2. respfig2:** 

2) Most likely, celsr1a protein is not being made, but the Western blot in Figure 4H lacks loading controls and makes interpretation difficult. Further, what is the predicted size of the protein, and how does it match with the multiple bands identified? Why is the major band in frnt Mc lane higher than the major band in wt Mc? Is this a cross-reactive protein in the Mc frnt sample, which is expected to not make viable celsr1a protein? I would support that the Western needs a loading control, and Materials and methods section needed describing/showing validation of the antibody used, if the Western is ultimately included.

In response to this concern of the reviewer, we have removed the Western from this figure. As stated in reviewer comments, the analysis does not really add anything given the number and nature of alleles with comparable phenotype which is clearer evidence of loss-of-function.

3) In principle, the knock-in of GFP to the celsr1a locus is an elegant way to monitor its expression, but the statement "The identified line, celsr1a^GFP^, recapitulates early expression seen by whole mount in situ (1dpf, Figure 5C)" requires a supporting in situ to the endogenous transcript in this line. I think to claim that the GFP knock-in reporter recapitulates the endogenous cels1ra expression pattern, an in situ to endogenous cels1ra (in situ probe published) is needed compared to reporter GFP (fine if separate sib embryo done by fluorescent microscopy, no need for GFP double in situ).

We have included in the figure a whole mount in situ of *celsr1a* expression at comparable early stages. Later stages the signal becomes faint. This is comparable to published in situs and also that found on ZFIN.

4) The use of label retention as a functional read-out for stem/progenitor cells in the gut is interesting, but why are Anterior BrdU cells/rugae increased at 48 hours of chase in frnt mutants relative to earlier time points? Even with decreased proliferation in the frnt mutants, should not all label-retaining populations stay constant or decreased over time? There may be variability in the degree of baseline labeling for different fish/replicates, but otherwise seems pretty consistent in other regions of the gut and at other time points in other conditions. A comment is sufficient in the Results/Discussion explaining the changes in label retention over time between the earliest and 48 hour time point for the mutant group. How is there increase in label retaining cells over time from 5 to 48 hours in the anterior segment in the cels1ra group? Technical issue? Perhaps different regions were assessed at the different time point?

We believe this is a technical issue and thank the reviewer for their insight in seeing this. In review of the data stemming from the reviewer comment, we noticed that the 48 hour time point may have fewer rugae than other time points. This seems to be limited to only that group as all the other controls were comparable. We revisited means of normalizing between individuals that may compensate for this, however analysis using both BrdU^+^ cells per/ total DAPI positive cells as well as overall area, and have found that Rugae is the most robust means of normalizing the data – as total cell # as well as area was too variable within a sample. This is probably due to proliferating cells being concentrated at the valleys between rugae. We will note the potential confounding effect of low rugae # in interpreting the data. This is now found in Materials and methods and in the legend of Figure 7—figure supplement 2.

[Editors' note: further revisions were suggested prior to acceptance, as described below.]

The manuscript has been improved but there are some remaining issues that need to be addressed before acceptance, as outlined below:In response to reviewer 1's concern 1, you added new data on intestinal goblet cell number (Figure 9K, L). The new data shows that the frnt genotype does not alter the observed increase in goblet cell number that occurs with CR. This is useful new data, but there appears to be no statistical analysis to support the claim that CR increases goblet cell number (for either genotype or intestinal region). Please add analysis of statistical significance to support this claim, or state in the legend that those differences did not reach statistical significance. Also, there is no panel M in this figure anymore, so please edit the figure legend accordingly.

We have added statistical analysis showing significant difference in goblet cell number after CR treatment in both middle and posterior regions. We have added notation of significance and added details to the figure legend. We thank the editor in noticing the leftover text from process of review and we have deleted the citation of panel M.